# A deep population reference panel of tandem repeat variation

Helyaneh Ziaei Jam[1], Yang Li[2], Ross DeVito [1], Nima Mousavi[3], Nichole Ma [2], Ibra Lujumba [4], Yagoub Adam [5], Mikhail Maksimov[1], Bonnie Huang [6], Egor Dolzhenko [7], Yunjiang Qiu [7], Fredrick Elishama Kakembo[4], Habi Joseph [4], Blessing Onyido [8,9], Jumoke Adeyemi [8,9], Mehrdad Bakhtiari[1], Jonghun Park [1], Sara Javadzadeh[1], Daudi Jjingo[4,10], Ezekiel Adebiyi[5,8,9,11], Vineet Bafna [1] & Melissa Gymrek [1,2] ✉

Tandem repeats (TRs) represent one of the largest sources of genetic variation in humans and are implicated in a range of phenotypes. Here we present a deep characterization of TR variation based on high coverage whole genome sequencing from 3550 diverse individuals from the 1000 Genomes Project and H3Africa cohorts. We develop a method, EnsembleTR, to integrate genotypes from four separate methods resulting in high-quality genotypes at more than 1.7 million TR loci. Our catalog reveals novel sequence features influencing TR heterozygosity, identifies population-specific trinucleotide expansions, and finds hundreds of novel eQTL signals. Finally, we generate a phased haplotype panel which can be used to impute most TRs from nearby single nucleotide polymorphisms (SNPs) with high accuracy. Overall, the TR genotypes and reference haplotype panel generated here will serve as valuable resources for future genome-wide and population-wide studies of TRs and their role in human phenotypes.

The availability of whole genome sequencing (WGS) datasets from thousands of individuals has enabled characterization of human genetic variation at unprecedented scale. Initial variant discovery efforts using low-coverage WGS were focused on single nucleotide polymorphisms (SNPs) and short insertions or deletions (indels)[1]. More recently, high-coverage WGS has enabled more accurate catalogs of short indels and structural variation[2]. Although multiple large WGS datasets now exist[2–4], variants in tandem repeat (TR) regions are largely underrepresented, in part because they require more specialized bioinformatics approaches.

Here we consider two types of TRs: short tandem repeats (STRs) consist of repeat units of 1–6 bp in tandem, whereas variable number tandem repeats (VNTRs) have longer repeat units. TRs experience rapid mutation rates that result in frequent changes in copy number[5]. Collectively, they comprise around 3% of the human genome[6] and occur at more than 2 million distinct loci[7,8]. TRs have been implicated in a variety of Mendelian disorders[9] and complex traits[10]. Although TRs represent one of the largest sources of human genetic variation, they are technically challenging to genotype, and are only partially captured by general SNP and indel genotyping tools used in standard variant calling pipelines.

[1]Department of Computer Science and Engineering, University of California San Diego, La Jolla, CA, USA. [2]Department of Medicine, University of California San Diego, La Jolla, CA, USA. [3]Department of Electrical and Computer Engineering, University of California San Diego, La Jolla, CA, USA. [4]The African Center of Excellence in Bioinformatics and Data Intensive Sciences, the Infectious Diseases Institute, Makerere University, Kampala, Uganda. [5]Covenant University Bioinformatics Research (CUBRe), Covenant University, Ota, Ogun 112233, Nigeria. [6]Department of Bioengineering, University of California San Diego, La Jolla, CA, USA. [7]Illumina Incorporated, San Diego, CA 92122, USA. [8]Department of Computer & Information Sciences, Covenant University, Ota, Ogun 112233, Nigeria. [9]Covenant Applied Informatics and Communication Africa Centre of Excellence (CApIC-ACE), Covenant University, Ota, Ogun 112233, Nigeria. [10]Department of Computer Science, Makerere University, Kampala, Uganda. [11]Applied Bioinformatics Division, German Cancer Research Center (DKFZ), Heidelberg, Baden-Württemberg 69120, Germany. ✉e-mail: mgymrek@ucsd.edu

Over the last decade, TR genotyping has rapidly matured. Variants, including large expansions, at STRs and VNTRs can now be reliably detected by multiple methods from high-coverage short read WGS[7,8,11–14]. These methods have been applied to catalog genome-wide TR variation across thousands of individuals from diverse populations. One of the earliest catalogs profiled nearly 700,000 STRs using low-coverage WGS from phase 1 of the 1000 Genomes Project (1000GP) cohort[15]. Subsequent studies have analyzed TR variation in deep WGS from other cohorts[4,10,16–20]. However, these have faced important limitations. Most available large WGS datasets have been biased toward European individuals. Those including more diverse populations were either low-coverage, resulting in low accuracy and high rates of missing genotypes, or had relatively small sample size.

Another important limitation of existing TR catalogs is that none provides a comprehensive view of TR variation. Most tools begin with a reference set of TRs based on a reference genome. However, reference sets vary dramatically across tools due to differences in parameters used to define repeats, which are often based on the limitations of individual genotyping approaches. For example, GangSTR[13] can identify large expansions but only genotypes TRs with no sequence imperfections, whereas imperfect repeats are considered by HipSTR[8]. ExpansionHunter[12] models imperfect repeats, but the reference set must be semi-manually defined by the user and may differ from that used by other tools. Further, the set of repeat unit lengths considered differs by tool (HipSTR considers 1–6 bp units, GangSTR 1-20 bp, adVNTR 6+bp). Thus, no single tool captures the full spectrum of TR variation.

Here, we develop a new method, EnsembleTR, which takes TR genotypes output by existing tools (currently ExpansionHunter, adVNTR, HipSTR, and GangSTR) as input, and outputs a consensus TR callset by converting TR genotypes to a consistent internal representation and using a voting-based scheme. We apply EnsembleTR to genotype 1.7 million TRs based on the hg38 reference genome across deep PCR-free WGS for 3202 individuals from the 1000GP[2] and PCR+ WGS data for 348 individuals from H3Africa Project[21]. We apply this resource to characterize population-specific TR variants, identify novel sequence-context features contributing to TR variability, identify TRs associated with gene expression, and generate an improved phased SNP-TR reference haplotype panel. The full set of phased genotypes as well as population-specific summary statistics are made publicly available to facilitate use by the genomics community. Overall, we envision this will be a powerful resource enabling the study of TR variation across a wide range of future applications.

## Results

### A genome-wide catalog of TR variation

We performed genome-wide genotyping of TRs using high-coverage PCR-free WGS data available for 3202 samples from the 1000 Genomes Project (1000GP) and PCR+ WGS data for 348 samples from the H3Africa Project (Methods). Both datasets were sequenced to an average of approximately 30x coverage. We applied four separate TR genotyping methods which consider a variety of TR classes, including short STRs (HipSTR, ExpansionHunter, GangSTR), STR expansions (GangSTR, ExpansionHunter), and VNTRs (GangSTR, adVNTR). All four methods take as input a reference set of TRs and output inferred diploid repeat lengths in each sample. HipSTR additionally identifies sequence differences between repeat alleles. Genotypes from each method were filtered to remove poor-quality calls (Methods).

We next developed a novel ensemble calling method (EnsembleTR) which takes as input VCF files from different TR genotypers and outputs a consensus set of genotypes (Fig. 1a). TRs genotyped by a single tool do not require merging and are simply added to the output. EnsembleTR then identifies overlapping TR regions genotyped by two or more tools, infers a mapping between alternate allele sets reported by each method, and outputs a consensus genotype and quality score for each call (Methods). We applied EnsembleTR to jointly genotype all samples, which resulted in consensus calls at 1,785,572 unique TRs (42% homopolymers) on autosomal chromosomes. After removing TRs called in fewer than 75% of samples, 1,714,353 TRs (41% homopolymers) remained. Of those, 55% were only genotyped by a single method (Fig. 1b), largely reflecting differences in TR reference sets published for each tool.

We evaluated whether the resulting consensus genotypes capture the expected patterns of genetic variation in our cohort. We first examined patterns of Mendelian inheritance (MI) in the 602 available trios (Methods). Overall, 94% of calls follow MI, and this rate increases with increasingly stringent score thresholds (Fig. 1c). TRs called by multiple methods typically show higher MI rates (Fig. 1b). Further, TRs called by HipSTR and ExpansionHunter have higher overall MI rates than TRs called by GangSTR and adVNTR (Fig. 1b).

Some improvement in MI at TRs genotyped by multiple methods could result from differences in TR composition across reference sets for each method, with TRs considered by multiple methods being potentially easier to genotype compared to those genotyped by a single method. To test this, we computed MI separately for each genotyper at calls on Chromosome 1 that are shared vs. unique across methods. adVNTR was excluded from this analysis since it genotypes a largely distinct set of TRs (Fig. 1b). For all methods, MI was highest at calls genotyped by all 3 methods compared to those called by a single method (Supplementary Data 1). This difference was most pronounced for GangSTR, which showed MI of 98.2% at shared calls vs. 88% for calls made by GangSTR alone.

Next, to determine whether ensemble-based calling improves genotype quality beyond what can be explained by differences in TR composition alone, we compared the MI of EnsembleTR calls vs. that obtained by individual methods at TRs on Chromosome 1 genotyped by multiple tools. For most combinations of methods, although in most cases the different genotypers were highly concordant (range 80-95%), we still observed that EnsembleTR calls either exhibited greater Mendelian consistency or achieved comparable MI to the genotyper with the highest MI level (Supplementary Data 2). For example, at 15,250,413 calls genotyped by both HipSTR and GangSTR, the EnsembleTR MI rate is 99.3%, compared to 99.1% for HipSTR and 98.1% for GangSTR. The only exception was for TRs called only by GangSTR and ExpansionHunter but not HipSTR, for which EnsembleTR calls showed lower MI than ExpansionHunter alone. However, these TRs make up a small percentage (approximately 0.1%) of the total callset, and typically are longer (mean 30 bp in hg38 vs. 19 bp for other TRs) making them more difficult to genotype precisely. We also observed that in 72% of calls that were genotyped by multiple methods but where at least one method did not follow MI, EnsembleTR calling resulted in Mendelian consistency, compared to 65% obtained by a naïve approach of always choosing the HipSTR genotype. Overall, our results suggest that while TRs called by multiple methods tend to be easier to genotype in general, EnsembleTR still improves call quality over any single method at these loci. For downstream analyses, we filtered calls from TRs with Mendelian error rates above 5% which resulted in 1,443,686 unique TRs.

To further evaluate our callset we performed fragment analysis via capillary electrophoresis to genotype 48 TRs on a subset of samples. Our validation panel includes 11 TRs implicated in repeat expansion disorders, plus an additional 37 TRs spanning a range of repeat classes. Each TR in our panel was tested on 48 samples, including 25 samples chosen to represent diverse population groups from the 1000GP, 17 additional samples from the "Platinum Genomes" multi-generational pedigree (6 of which are included as 1000GP trios), and 6 samples from the Genome In a Bottle project (Supplementary Data 3–7; Supplementary Figs. 1-2; Methods). Out of 1395 mutual calls between EnsembleTR and fragment analysis, 1362 (98%) were concordant (96% concordance for GangSTR and 98% for HipSTR calls). Of the 33

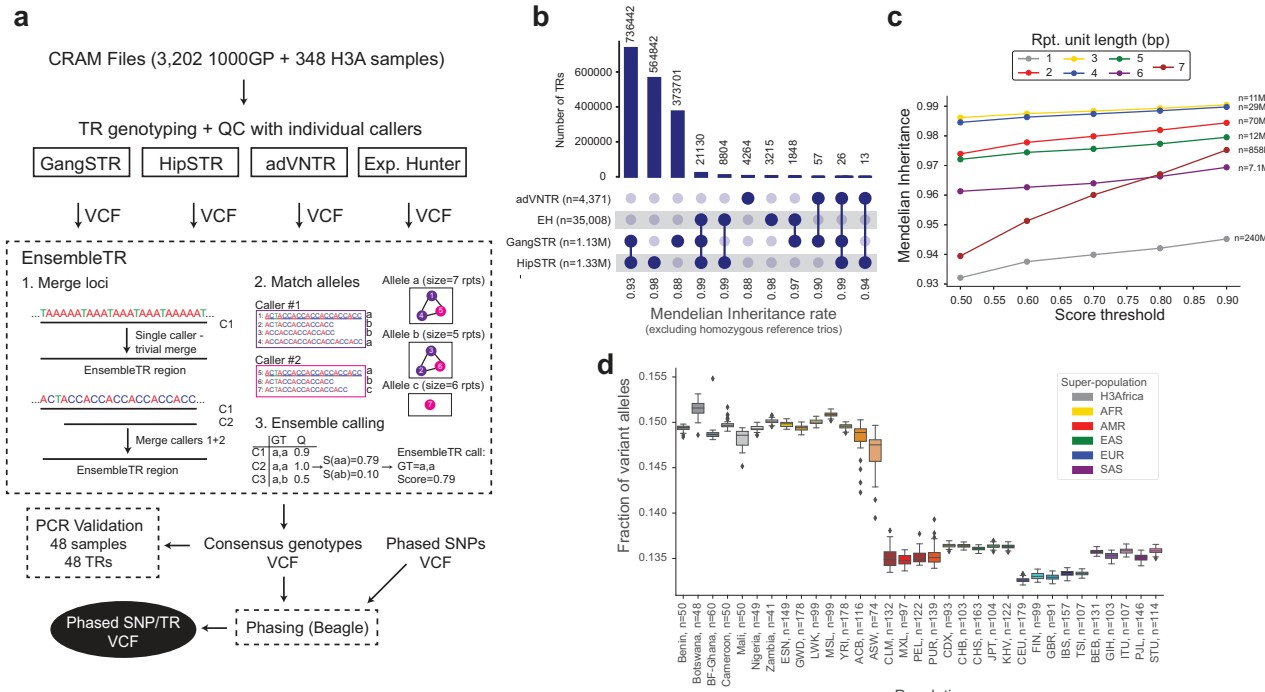

**Fig. 1 | A deep catalog of TR variation across human populations. a** Overview of EnsembleTR workflow. Aligned reads are input to four TR genotyping tools (GangSTR, HipSTR, adVNTR, and ExpansionHunter). Filtered VCFs are input to EnsembleTR. EnsembleTR first identifies sets of mergeable loci (step 1) and identifies sets of compatible alleles between callers (step 2). Finally, it scores each possible diploid genotype (step 3) and outputs the best genotype and its score. The resulting VCF file is used to generate a phased SNP + TR reference haplotype panel. **b** Overlap of TRs called by each method. Annotations below the bars indicate the combination of methods a TR was called in. Numbers next to each method indicate the number of unique TRs in each category. Numbers below the plot indicate the Mendelian Inheritance (MI) rate across all calls in each category. Categories with fewer than 10 total TRs were excluded. **c** Mendelian Inheritance as a function of EnsembleTR quality score. The *x*-axis gives the EnsembleTR quality score threshold used, and the *y*-axis gives the percent of genotyped trios that follow MI. Line colors denote repeat unit lengths. Each trio/TR pair was only included in each category if all calls in the trio passed the score threshold. Trio/TR pairs for which all samples were homozygous for the reference allele were excluded from analysis. **d** Distribution of the fraction of non-reference alleles in individuals by population. Boxplots summarize the distribution of the fraction of variant alleles in each sample. Horizontal lines show median values, boxes span from the 25th percentile (Q1) to the 75th percentile (Q3). Whiskers extend to Q1−1.5*IQR (bottom) and Q3+1.5*IQR (top), where IQR gives the interquartile range (Q3-Q1). Homopolymer TRs are excluded. Box colors denote superpopulations. Gray denotes H3Africa. Other colors denote 1000 Genomes superpopulations.

discordant calls, 9 were from a single TR (C9orf72), a GC-pure repeat (which comprises 0.67% of our catalog). Nearly all errors including those at C9orf72 resulted either from discrepancies of a single unit or by dropout of one of the alleles at a heterozygous locus in either technology. Notably, our validation focused on TRs that could be readily genotyped by PCR, and thus excluded more complex repeats as well as homopolymer loci, for which error rates are likely higher. Still, our results suggest that most non-homopolymer TR genotypes in our catalog based on WGS are of comparable accuracy to those obtained by the experimental gold standard of fragment analysis.

Next, we examined population-specific allele frequencies at well-characterized TRs, including known pathogenic loci and those used for forensics analysis, and found that EnsembleTR results recapitulated published results for these loci (Methods; Supplementary Figs. 3, 4), with mean Jensen-Shannon divergence ranging between 0.03-0.21 for forensic loci and 0.1-0.31 for pathogenic loci across all super-populations (compared to mean 0.43 and 0.55 when permuting loci; Methods). We then examined genome-wide patterns of TR variation across populations. Initial inspection of the number of variant TR alleles per sample showed that H3Africa samples had far higher rates of polymorphism even compared to African samples in the 1000GP (Supplementary Fig. 5). However, we hypothesized this could be driven by the PCR+ nature of the H3Africa samples, which induces high error rates, in particular at homopolymer TRs[22]. Repeating this analysis, but excluding homopolymer loci, revealed similar patterns as were observed for other classes of variants[2] (Fig. 1d). Individuals from African populations had the highest number of variant TR alleles

compared to the reference, whereas Europeans had the fewest. Further, admixed African individuals showed the highest variability in the number of variants per sample. As expected, the rate of discovery of new TR alleles slows with each new individual, but this rate increases after the addition of African samples and continues to increase when H3Africa samples are added (Supplementary Fig. 6). Performing principal component analysis (PCA) on a matrix of the sum of repeat lengths at each TR for each sample captured expected patterns of population structure (Supplementary Fig. 7).

## Characterizing population-specific TR variation

We next characterized patterns of TR variation and how they vary across populations. After filtering, we identified an average of 184,056 and 184,759 TRs in each sample for which one or both alleles did not match the reference genome, respectively (Supplementary Data 8). Our callset contains 6758 TRs entirely inside coding exons, corresponding to 0.5% of TRs genotyped genome-wide (Supplementary Data 9). On average, each sample contained at least one non-reference allele at 295 coding TRs. As expected, TRs with repeat unit lengths that are multiples of 3 are over-represented in coding exons whereas mononucleotide, dinucleotide, and tetranucleotide TRs are far more prevalent in non-coding regions of the genome (Supplementary Fig. 8a). Additionally, a far lower percentage of TRs in coding regions are polymorphic (51% for coding exons compared to 78% genome-wide; Supplementary Fig. 8b).

We then summarized the variability in the length of each TR by computing the heterozygosity (H) (and counting the number of alleles

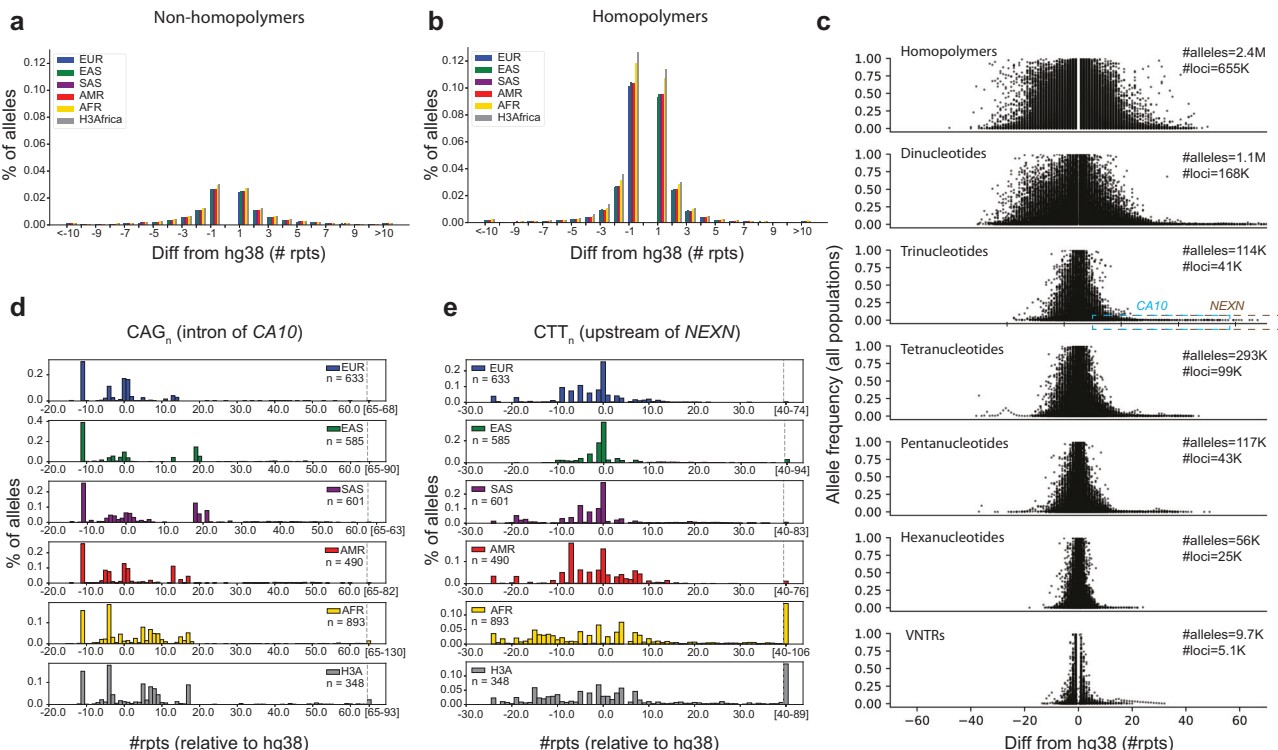

**Fig. 2 | Characterizing population-specific TR variation. a, b** Distribution of variant allele sizes. Bars show the percent of variant alleles that have a specified difference in length compared to the hg38 reference. Positive numbers indicate expansions and negative numbers indicate contractions relative to the reference. Panel (**a**) shows data for all non-homopolymer TRs and (**b**) shows data for homopolymer TRs. Bar colors denote superpopulations. Gray denotes H3Africa. Other colors denote 1000 Genomes superpopulations. **c** Allele frequency vs. allele length. The *x*-axis shows allele lengths relative to the reference genome and the *y*-axis shows the frequency of each allele across all populations. Different panels denote different repeat unit lengths. Dots corresponding to expansion alleles highlighted in the text are annotated with dashed boxes. Only alleles with frequency at least 0.1% are shown. Alleles with the same length as the reference allele are excluded. **d, e** Population-specific allele distributions at example loci. In each panel, the *x*-axis denotes allele length (number of repeats relative to hg38) and the *y*-axis denotes the frequency of each allele. Each panel shows a different superpopulation. Panel (**d**) shows an intronic trinucleotide repeat in *CA10*. Panel (**e**) shows a trinucleotide repeat upstream of *NEXN*. Both repeats have expansion alleles common in African populations compared to non-Africans.

with frequency ≥ 1%; Supplementary Fig. 9; Methods). TRs in our catalog show a wide range of polymorphism rates, with 56% of non-homopolymer TRs (6% of homopolymers) fixed or nearly fixed at a single allele length (H < 0.001), 11% (29% of homopolymers) with two common alleles, and 17% (50% of homopolymers) with three or more common alleles. TR heterozygosity and the number of common alleles are highly correlated across populations (Supplementary Figs. 10-11), with few TRs being polymorphic in one population but not others.

The majority of alleles identified in each sample differ in length from the reference genome by only a small number of repeat units, and this trend is consistent across populations (Fig. 2a, b). All populations show a slight bias toward alleles that are shorter than the reference allele, and exhibit the highest rates of variation at homopolymer TRs. Overall, alleles closest in length to the reference allele tend to be common, whereas alleles tend to decrease in frequency as a function of their length difference from the reference (Fig. 2c). However, we observed 196 TRs at which more than 95% of observed alleles differed from the reference by more than 2 repeats (Supplementary Data 10). Many of these consisted of highly imperfect repeats or TRs with multiple distinct repeat units. We manually inspected available PacBio "HiFi" reads from a single sample (Methods) overlapping each of these TRs, and found that the EnsembleTR allele was supported at 194/196 loci (examples shown in Supplementary Fig. 12). We further investigated these in the new T2T reference genome[23] and found that for 194/195 TRs that could be successfully lifted over, the T2T reference matched the most common allele called by EnsembleTR. Overall, this suggests that a subset of complex TRs may not be correctly represented in the hg38 reference but are resolved in T2T.

We also observed a subset of common alleles with large expansions compared to the reference (Fig. 2c). To identify population-specific polymorphic repeat expansions, we searched for TRs with common expansions in either Africans or non-Africans but not both. We filtered homopolymer TRs and only considered expansions as outlier alleles with copy number >10 (Methods). This method identified 264 candidate TRs (Supplementary Data 11). Of these, 198 were specifically expanded in Africans and 66 in non-Africans. We additionally applied two methods specifically designed to detect pathogenic repeat expansions (STRetch[24] and ExpansionHunter Denovo[11]) to identify candidate expansions in the H3Africa cohort. Of the 198 candidate TRs, 11 were supported by at least one and 5 were supported by both methods. Two TRs had particularly dramatic Africa-specific expansion alleles (Fig. 2d, e), both of which were supported by STRetch and ExpansionHunter Denovo. These include an intronic CAG repeat in *CA10* (1.6% of African alleles have >+65 copies relative to hg38 compared to 0.13% in non-Africans, originally genotyped by ExpansionHunter, HipSTR, and GangSTR) and a CTT repeat upstream of *NEXN* (14% of African alleles have >+39.7 copies relative to hg38 compared to 1.2% in non-Africans, originally genotyped by ExpansionHunter and HipSTR). Notably, previous studies of the *CA10* locus reported expansions at up to 30% of alleles[25], far more than reported for any population here. While we still likely underestimate expansion rates (see below), we note these studies used a threshold of 40 total copies to define expansions, compared to 86 total copies here (+65 relative to the 21 copies in hg38) based on our outlier detection approach.

To further validate the *CA10* and *NEXN* expansions, we compared EnsembleTR calls to genotypes obtained by manual inspection of

PacBio HiFi reads available for 27 1000GP samples (Supplementary Data 12). Notably, long alleles at both repeats are much longer than the Illumina read length, and so repeat length estimates are inexact. Still, allele lengths estimated by EnsembleTR are strongly correlated with length estimates based on HiFi reads (Pearson $r = 0.78/0.89$, two-sided $p = 1.5e$-6/3.6e-10 for *CA10* and *NEXN* respectively). Further, all large expansion alleles identified by EnsembleTR were supported by PacBio, although some large expansions were missed as a result of a bias in EnsembleTR's voting scheme which down-weights lower confidence alleles (see Discussion). We additionally performed PCR amplification of the *CA10* repeat in four 1000GP samples with a range of allele lengths, which confirmed EnsembleTR genotypes including a large expansion at this locus (Supplementary Fig. 13). Interestingly, manual inspection of both TRs in HiFi reads revealed common variation not only in TR length, but also in TR sequence. At the *CA10* TR, which is annotated as a CAG repeat in hg38, most alleles are perfect CAG repeats but expansions of CCG or CGG were also observed (Supplementary Fig. 13c). Similarly, at the *NEXN* TR, which is annotated as a CTT repeat, many observed alleles instead consist of repeats of the hexamer sequence CTTCTC. This alternate repeat unit was observed on both expanded and normal range allele lengths.

Finally, we assessed whether these sequence imperfections are captured by HipSTR, the only method used here that attempts to infer the sequence as well as the length of TR alleles. We used TRViz[26] to visualize sequence structure for alleles at the *CA10* and *NEXN* TRs (Supplementary Fig. 14). HipSTR successfully identified the CTTCTC motif in shorter alleles at *NEXN* as well as CAC and CAT motifs at the *CA10* locus. Longer alleles could only be genotyped by Expansion-Hunter or GangSTR, which report perfect repeats even for genotypes where manual inspection of PacBio reads confirmed the presence of alternate motifs. Notably, EnsembleTR output annotates the method of origin of each call, which could be used to determine in which cases repeat allele sequence information is reliable.

## Sequence determinants of TR heterozygosity

We next used our catalog to examine determinants of polymorphism patterns across different TRs by correlating sequence features with TR heterozygosity. As widely observed previously[15,27,28], we found that TR heterozygosity is most strongly correlated with total repeat length and the length of the repeat unit (Fig. 3a), with TRs with longer total stretches of uninterrupted repeat sequence and shorter repeat units being typically the most polymorphic. This trend is consistently observed across populations (Supplementary Fig. 15). Among TRs in our catalog with the same repeat unit length, heterozygosity also varies to a lesser extent across different repeat unit sequences (Fig. 3b–e). For example, CG and AT dinucleotide repeats have higher average heterozygosities at a given length compared to AG or AC repeats. For tetranucleotides and pentanucleotides, AGAT and AAAAG repeats tend to have the highest heterozygosities across a range of repeat lengths. When visualizing reference TR length in bp vs. abundance in the genome, we additionally observed an unexpected periodic pattern for multiple repeat classes. Trinucleotides with length 0 mod 3 are less abundant than those consisting of a non-integer number of total repeat copies, similar to a previous observation[29]. Similarly, dinucleotides with an even total length (e.g. ACACAC) tend to be slightly less abundant than those with an odd total length (e.g. ACACACA). Similar periodic trends were observed for other repeat unit lengths (Fig. 3a).

Beyond the sequence of the TR, we reasoned that features of the genomic sequence flanking a TR may also impact its variability. To investigate this further, we focused on non-'GC' dinucleotide STRs in our catalog with repeat units AC/GT, AT, or AG/CT with reference length between 12 and 17 bp. We first classified TRs as either "stable" (major allele frequency = 1) or "polymorphic" (major allele frequency <0.99), resulting in a set of 3435 polymorphic STRs and 6922 stable

STRs (Fig. 3f). Of these 4829 had an AC/GT repeat unit, 3051 had AG/CT, and 2477 had AT. We then applied two methods to identify sequence features characteristic of stable TRs. First, we applied HOMER[30], a motif discovery tool, to sequences extracted from a 64 bp window on each side of the TRs of each repeat unit separately. For AC repeats, HOMER identified five motifs enriched in the context of polymorphic vs. stable TRs. Of these, four contain a repetitive motif with a dinucleotide repeat unit (Fig. 3g). Similar top motifs were identified for AT, but not AG/CT repeats (Supplementary Fig. 16).

Second, we trained a convolutional neural network (CNN) using 58 bp flanking each TR plus 6 directly adjacent bases that make up the TR from both sides, and used gradient-based attribution scores to quantify the importance of each input base (Methods). Our model achieved an overall accuracy of 73% on a held out test set with area under the precision recall curve 0.82/0.62 when considering stable/polymorphic STRs as the target class (precision = 0.76/0.64 for stable/polymorphic and recall = 0.87/0.46 for stable/polymorphic). Visualization of attribution scores for the TRs most confidently and correctly predicted to be polymorphic identified that nearby dinucleotide repeat-like sequences have the strongest influence on whether the model predicted an STR to be polymorphic (Fig. 3h). This pattern was not visible in TRs confidently correctly predicted to be stable (Supplementary Fig. 17). This result is consistent with our HOMER findings that dinucleotide repeat-like sequences in the flanking regions of dinucleotide TRs results in increased heterozygosity.

To validate these results and quantify the strength of the relationship, we found the count of all 4-mers composed of adjacent dinucleotide or mononucleotide motifs (e.g. ATAT, ACAC, AAAA, etc.) in a 64 bp window around all dinucleotide STRs. When dividing the counts of these repetitive 4-mers in context regions into quintiles, polymorphic STRs are highly prevalent in the upper quintile (27.6% of polymorphic STRs fall into this bin, compared to 15.5% of stable STRs) and depleted in the lowest quintile (which contains 16.1% of polymorphic STRs and 23.1% of stable STRs). Overall, the count of these context 4-mers is significantly correlated with STR heterozygosity both overall and individually for STRs with AT, AC/GT, and AG/CT repeat units (Bonferroni corrected $P$-values $1.5 \times 10^{-124}$, $9.7 \times 10^{-10}$, $5.3 \times 10^{-30}$, and $1.0 \times 10^{-5}$). Although these context dinucleotides are predictive, in all cases the strength of the correlation with these sequence context features is less than the correlation with copy number (Fig. 3i; Supplementary Data 13).

## Detecting TRs associated with gene expression

To assess the utility of our catalog in identifying trait-associated TRs, we performed expression quantitative trait loci (eQTL) discovery in 452 unrelated samples (363 EUR and 89 AFR) with available RNA-sequencing derived from lymphoblastoid cell lines (LCLs) from the Geuvadis project[31]. We tested for association between the sum of the repeat length of the alleles of each individual and gene expression for each TR within 100 kb of each gene (Methods; Supplementary Data 14-15; Data Availability; Fig. 4a). Tests were performed separately in the EUR and AFR cohorts. In total, we identified 74,340 (EUR) and 400 (AFR) individual significant TR-gene pairs (FDR < 0.05) and 3664 (EUR) and 81 (AFR) total eGenes (gene-level FDR < 0.05). Effect sizes of eTRs significant in at least one cohort (EUR or AFR) were strongly correlated across these two cohorts (Pearson $r = 0.40$; $p < 10^{-200}$; $n = 64,760$; Fig. 4b). For comparison, a previous eTR analysis we performed in this cohort[32] based on low-coverage WGS from the 1000GP identified only 2060 eGenes at the same FDR threshold.

We compared eTR effects measured here to eSTRs we identified previously across 17 tissues in the Genotype-Tissue Expression (GTEx) cohort[16]. Effect sizes computed for overlapping sets of TR-gene pairs across studies were significantly correlated in all tissues ($p < 10^{-200}$ and $p = 5.7e-12$ in all tissues considering eTRs significant in European and African Geuvadis cohorts, respectively) but were most strongly

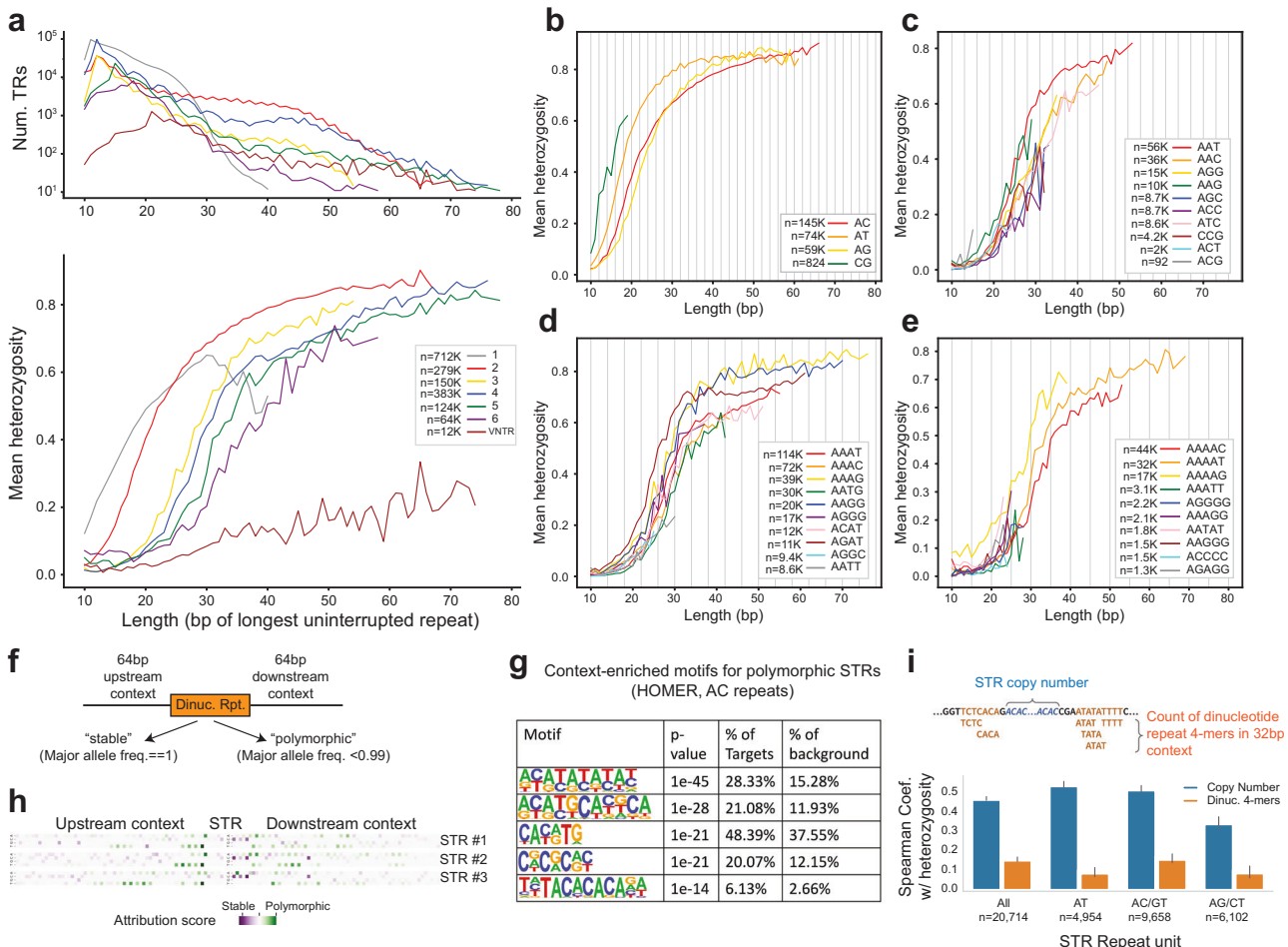

**Fig. 3 | Sequence determinants of TR polymorphism. a** Heterozygosity is correlated with total TR length. The *x*-axis denotes the length of each TR in hg38 (bp of the longest perfect repeat). The top panel gives the number of repeats in each category (repeat unit length in bp). The bottom panel shows the mean heterozygosity for TRs with each length. **b**–**e** are the same as the bottom panel of (**a**), except for different repeat unit sequences. Homopolymers are not shown separately as most have the same repeat unit (A$_n$). Gray lines are shown every other bp in b, every third bp in c, every fourth bp in d, and every fifth bp in (**e**). **f** Overview of approach to classify TRs as stable vs. polymorphic based on sequence context. We used two approaches (HOMER and convolutional neural networks) to classify dinucleotide TRs based on 64 bp of sequence context upstream and downstream of the TR. **g** Top motifs enriched in the context of AC dinucleotide TRs. All other discovered motifs were flagged as likely false positives by HOMER. One-sided unadjusted binomial *P*-values are shown. **h** Attribution scores of three example AC TRs predicted to be polymorphic. Each row denotes a different TR. Within each row, the matrix has a row for each nucleotide and a column for each position (centered on the TR). Color denotes the attribution score of each base in each position (green and purple denote positive and negative contributions toward polymorphism prediction, respectively). **i** Correlation of TR and context features with heterozygosity. Blue bars denote the Spearman correlation of TR length (reference copy number) with heterozygosity. Orange denotes correlation of the counts of dinucleotide-like or homopolymer-like 4-mers in the context region (±64 bp) with heterozygosity. Error bars give 99% confidence intervals around the estimated Spearman correlation found by bootstrapping with 1000 70% subsets.

correlated with Cultured Fibroblasts (Supplementary Data 16, Fig. 4c, d). The previous GTEx analysis excluded LCLs due to low sample size, and so we could not directly compare to data from the same cell type. eQTLs discovered here recapitulate known signals, and also identify novel trait-associated TRs. For example, one of our top eTR signals is a dodecamer repeat in the promoter of *CSTB*, which has been reported by multiple previous studies[16,33] and is associated with myoclonus epilepsy[34] (Fig. 4e). We identified a total of 2996 eTRs significant at the gene-level that were either not previously tested (2917) or did not reach at least nominal significance (79) in any tissue tested in GTEx. An example novel association of a dinucleotide AT repeat with *TIMM10* expression is shown in Fig. 4f.

## Phased reference panel allows accurate imputation of TR variants

Finally, we generated a phased reference haplotype panel of SNPs/short indels and TRs from the 1000GP samples. We used our previously published pipeline[35] to phase each TR separately onto a backbone of phased SNPs in a ±50kb window, resulting in a single panel containing both phased SNPs and TRs (Methods). The resulting panel contains a total of 1,091,550 TRs, compared to 453,671 TRs in the previously published panel. We assessed the utility of this panel for imputing TRs by performing a leave-one-out analysis at TRs on chromosome 21 and observed an average concordance of 99% between imputed genotypes and observed genotypes in all 5 superpopulations of 1000GP. For comparison, we performed a naive imputation method in which each genotype is imputed as the most common diploid genotype, which resulted in an average concordance of 87%. As expected, imputation performance is strongest at the least polymorphic TRs, and weakest at those that are highly multi-allelic and/or have the highest heterozygosity (Supplementary Fig. 18; Fig. 5a).

We then evaluated the effectiveness of our panel in imputing medically relevant TRs. Utilizing our panel, we imputed genotypes for 92 repeats, comprising 16 widely recognized pathogenic loci involved in expansion disorders as well as 76 potential causal variants associated with blood traits that were included in our panel

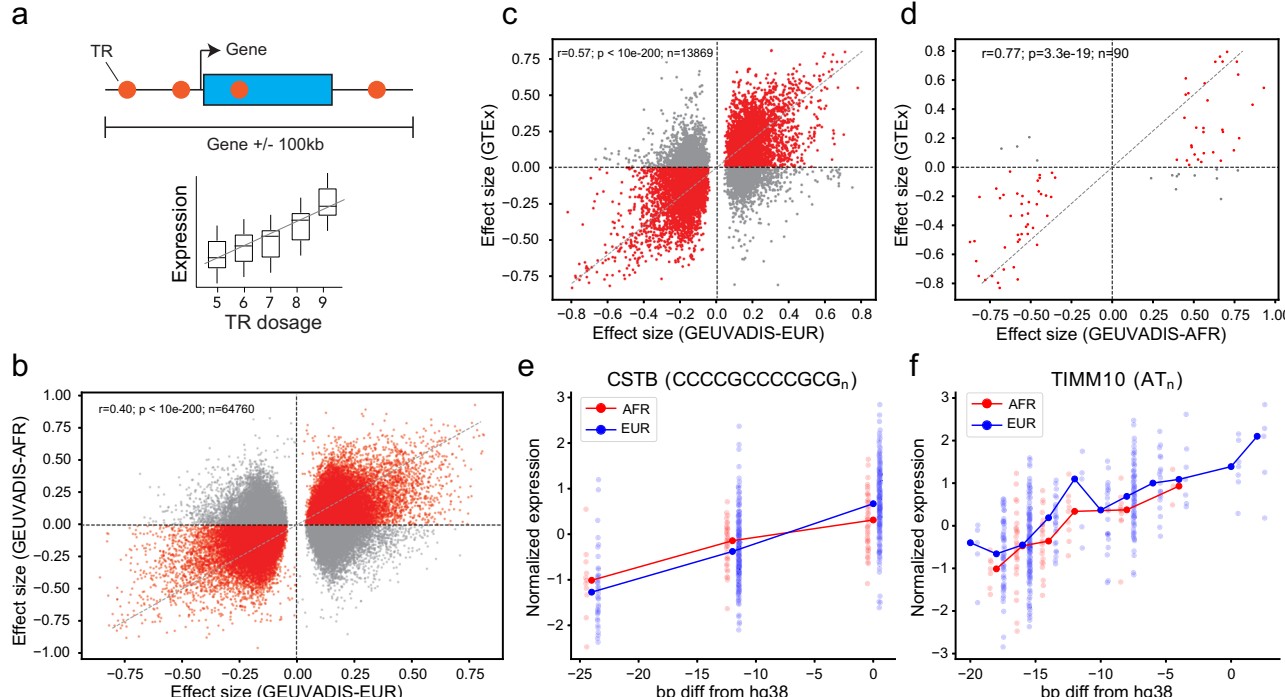

**Fig. 4 | TRs associated with gene expression in LCLs. a** Schematic overview of eTR detection. A separate association test between TR dosage (sum of repeat lengths) and expression is performed for each TR within 100 kb of a gene. **b** Comparison of effect sizes across populations. The *x*-axis gives effect sizes based on European samples and the *y*-axis gives effect sizes based on African samples from GEUVADIS. Each dot represents a TR-gene pair (eTR). eTRs with consistent effect directions are colored in red. Only eTRs reaching FDR < 0.05 in at least one population are included. r gives the Pearson correlation between effect sizes. The P-value is two-sided and not adjusted for multiple comparisons. **c**, **d** Comparison of effect sizes in GEUVADIS vs. GTEx. The *x*-axis gives effect sizes measured in GEUVADIS in Europeans (**c**) or Africans (**d**). The *y*-axis of each plot gives the effect sizes measured in Fotsing et al. [16] in cultured fibroblasts. Each dot represents a TR-gene pair (eTR). Only eTRs with adjusted *P*-values < 0.05 in the GEUVADIS analysis are shown. r gives Pearson correlation between effect sizes. The *P*-value is two-sided and not adjusted for multiple comparisons. **e** Example replication of a previously identified eTR. The *x*-axis gives the number of repeats of a TR upstream of the gene *CSTB*. The *y*-axis gives normalized *CSTB* expression. **f** Example novel eTR. The *x*-axis gives the number of repeats of a TR near *TIMM10*. The y-axis gives normalized *TIMM10* expression.

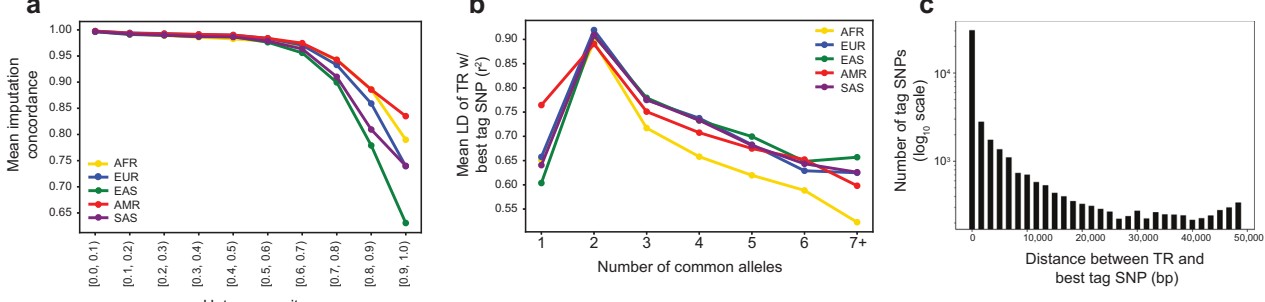

**Fig. 5 | Phasing and imputation at TRs. a** Imputation accuracy decreases with heterozygosity. The *x*-axis denotes TR heterozygosity. The *y*-axis denotes the mean concordance for TRs in each heterozygosity bin based on a Leave-One-Out analysis on chromosome 21. **b** TRs are often tagged by common SNPs. The *x*-axis denotes the number of common alleles (frequency > 0.01) for each TR. The *y*-axis denotes the mean LD ($r^2$) of the best tag SNP for TRs in each bin. For (**a**, **b**), colors denote 1000 Genomes superpopulation. **c** Distribution of the distance between each TR and its best tag SNP. The *y*-axis is given on a $\log_{10}$ scale.

(Supplementary Data 17). Notably, our panel is comprised of healthy controls, and thus imputation is restricted to common normal range alleles for the known pathogenic loci. Our analysis revealed an average concordance range 0.95–0.97 across populations for all examined loci. Performance was lower at known pathogenic TRs, with highly polymorphic TRs implicated in Huntington's Disease and spinocerebellar ataxia 3 being most challenging (concordance range 0.68–0.90). Overall the average concordance across all pathogenic loci remained high (range 0.89–0.92 across populations) indicating that common, non-pathogenic alleles at a range of medically relevant repeats can be accurately imputed.

Multiple TRs have recently been implicated as causal drivers of genotype-phenotype associations discovered using genome-wide association studies (GWAS)[10,36]. In these cases, although the TRs are likely causal, the signals were originally identified using nearby tagging SNPs in at least moderate linkage disequilibrium (LD) with the TR. We used our haplotype panel to explore the ability of nearby SNPs or small indels to tag TRs. We determined the best tag SNP for each TR as the SNP/indel within a ± 50 kb window with the strongest LD (Supplementary Data 18). As expected, TRs that are largely bi-allelic are often well-tagged by nearby SNPs (mean best tag SNP $r^2 = 0.90$), whereas the LD of the best tag SNP decreases for TRs with an increasing number of

common alleles (Fig. 5b). For example, for TRs with five common alleles the mean $r^2$ of the best tag SNP in Europeans is 0.68, with generally weaker tagging in the African superpopulation. The majority of tag SNPs are located within a small window around the TR, and in some cases the top tag SNP is within or directly adjacent to the TR (Fig. 5c). Overall, these results indicate that while bi-allelic TRs are likely well captured by existing variant panels used by GWAS and other studies, more polymorphic TRs are often not well tagged by a single nearby SNP or indel.

## Discussion

TRs represent some of the most polymorphic regions of the genome, but have so far not been systematically included in large genetic variation databases, in large part due to technical challenges in genotyping as well as discrepancies in how TRs are defined by different tools. Here, we developed a novel framework, EnsembleTR, which uses an ensemble approach to integrate the output of multiple TR genotypers and generate a deep catalog of TR variation in the 1000GP and H3Africa cohorts. After filtering low-quality calls, our ensemble approach genotyped more than 1.7 million TR loci. We applied EnsembleTR to identify population-specific repeat expansions, characterize sequence determinants of TR stability, perform eQTL analysis, and generate a phased TR-SNP reference haplotype panel.

Our TR catalog based on EnsembleTR contains substantially more TRs than can be genotyped with any single method. The largest gain comes from the fact that recommended reference TR sets have low overlap across tools (Fig. 1b), with many TRs included only by a single method. This lack of overlap stems from the algorithm design of each tool. For example, whereas HipSTR can genotype TRs with sequence imperfections, GangSTR is restricted to perfect repeats. On the other hand, adVNTR is designed specifically for VNTRs and not STRs. Beyond improving the quantity of TRs called, ensemble-based genotyping also increases genotype quality at TRs called by more than one method. For example, at TRs called by multiple methods in almost all cases EnsembleTR genotypes showed Mendelian consistency levels that were either better than any individual method or comparable to the best-performing method (Supplementary Data 2).

The TR dataset presented here provides important improvements over previous population-wide TR panels and their applications. Previous TR panels were primarily based on hg19 and on a single TR genotyper[4,15,37], or are under controlled access restrictions and have limited representation of diverse ancestries[16,18,20,36]. In contrast, this dataset is made freely available, is based on the hg38 reference genome, and integrates TR calls from four different tools. We also improve upon published efforts to identify TRs acting as eQTLs. We previously identified eTRs using low-coverage Phase 3 1000GP data[32], which was underpowered due to low TR genotyping quality. A separate study performed eTR analysis based on targeted sequencing of promoter TRs but focused on TRs within 1 kb of transcription start sites[38], therefore missing the majority of TRs. More recently, we analyzed eSTRs[16] and eVNTRs[17] in the GTEx dataset, but excluded LCLs in the STR analysis due to low sample number. The current study shows high concordance with these previous eTR efforts, but identifies 2996 novel TR-gene associations in LCLs. Finally, we present a new TR-SNP reference haplotype panel, with 1,091,550 loci compared to our previous panel of 453,671 TRs[35]. This panel can be used for imputing TRs into external GWAS datasets, an approach which has already proven successful by us and others to identify novel trait-associated STRs[36] and VNTRs[10].

While overall patterns of TR variation are highly similar across populations, detailed analysis of individual TRs revealed individual loci with population-specific patterns. For example, we identified multiple Africa-specific repeat expansions, including common trinucleotide expansions in an intron of *CA10* and in the promoter of *NEXN* which often involve expansions containing multiple distinct repeat units. These expansions are supported by both the African cohorts within 1000GP as well as within H3Africa. Common expansions of the repeat in *CA10*, a brain-expressed gene, have been previously reported (previously referred to as the ERDA1 locus[39]), and have been speculated to be associated with psychiatric disorders[40]. *NEXN* mutations have previously been shown to result in dilated cardiomyopathy[41], which is particularly prevalent in Africa[42]. Because 1000GP and H3Africa do not have phenotype information available, future efforts are needed to determine the potential phenotypic impacts of these population-specific expansions.

This study faced several limitations, many of which will be overcome as sequencing technology and genotyping algorithms continue to improve. First, our TR catalog is based on genotypes obtained from short reads. While this enables reliable genotyping of nearly 2 million TRs, including TRs longer than Illumina read lengths, it is far from comprehensive and many long and complex TRs are still missing. In particular, long alleles with complex structures consisting of multiple repeat units, such as pathogenic alleles in *RFC1* linked to CANVAS[43], or the imperfect repeats at long alleles at the *NEXN* and *CA10* repeats described above, are still challenging to resolve with short reads. Long reads such as PacBio HiFi show great promise to genotype the majority of these loci. Intriguingly, the new TRGT method[44] can reliably infer both repeat length and sequence structure even at complex repeats such as *RFC1*. However, PacBio data is currently only available for several dozen 1000GP samples. Second, while the 1000GP data is PCR-free, WGS from H3Africa is PCR+, likely resulting in high error rates in particular at homopolymer TRs and preventing reliable assessment of repeat expansions specific to that cohort. Third, several technical improvements to the EnsembleTR pipeline may improve future genotyping efforts. For example, it currently only merges TR records from two or more methods if the repeat unit is determined to be identical for that locus across all methods. This occasionally fails, for example at the *CSTB* promoter TR which is called as a 5-mer by HipSTR but a 12-mer by adVNTR. Further, EnsembleTR currently prioritizes allele lengths that can be most precisely estimated, which in some cases such as the *CA10* TR results in incorrectly choosing high-confidence HipSTR calls over large but inexact expansions identified by ExpansionHunter. Another future improvement is to incorporate ensemble-calling of TRs into pangenome-based methods, which have resulted in important improvements to variant calling at other variant types but are still not optimized for TRs[45].

Overall, this study presents a dataset of 1.7 million TRs across 3550 diverse individuals, as well as a phased TR-SNP reference haplotype panel. These calls are made publicly available (see Data Availability) and will serve as an important resource for future efforts to identify population-wide patterns of TR variation and study the effect of genetic variation at TRs on human phenotypes.

## Methods

### Dataset description

Whole genome sequencing CRAM files for 3202 1000GP samples, including 1598 males and 1604 females, aligned to GRCh38 were obtained from European Nucleotide Archive accessions PRJEB31736 (unrelated samples) and PRJEB36890 (related samples). Population and superpopulation labels for each sample were obtained from the 1000GP data portal (https://www.internationalgenome.org/data-portal/sample). As described on the 1000GP data portal (https://www.internationalgenome.org/1000-genomes-summary/), all collections included in the 1000GP Project followed their ethical guidelines and model informed consent language.

CRAM files for 348 H3Africa samples are available from the European Genome-Phenome Archive with accession code EGAS 00001002976 and were generated as part of the H3AChip Design project. The samples in the H3Africa dataset represent individuals from West, Central, and South African countries. Gender information for samples in the H3Africa cohort was not available to the authors.

CRAM files were accessed through the H3Africa Genome Analysis Working Group. As previously described in the original paper[21], all samples were collected after appropriate approvals had been obtained from local Ethics Boards and Committees in each of the represented countries, and participants gave informed consent.

Information on age of samples was not available for any of the cohorts. Gender of samples was not used in any of the downstream analyses unless otherwise specified.

## TR genotyping with published tools
We first used each tool (HipSTR, GangSTR, adVNTR, ExpansionHunter) to genotype TRs and generate raw calls in VCF format, with a single VCF file per population.

**GangSTR.** GangSTR[13] v2.4.5 was run on each sample separately with non-default parameters --str-info str_info_file (see below), --bam-samps sample_id, --samp-sex sample_sex, and --grid-threshold 250. We generated an initial set of reference TRs for the hg38 assembly using Tandem Repeats Finder[46] with the following parameters: match=2, mismatch=5, indel=17, maxperiod=20, pm=80, pi=10 and min-score=24. We then refined the reference set by applying a series of filtering steps. First, we removed repeats longer than 1 kb. Then, we kept a single repeat with the shortest repeat unit length among those with identical start or stop coordinates. Compound and imperfect repeats were removed and any extra bases not matching the repeat motif were trimmed from both sides. Any duplicated repeats were discarded post-trimming. We then removed any repeats from the reference that did not have a minimum number of 10, 5, 4, and 3 copies for homopolymers, di-, tri-, and tetra/penta/hexa-nucleotide repeats respectively. Finally, we filtered out any overlapping repeats if their motifs consisted of identical nucleotide types. The reference set is available at https://s3.amazonaws.com/gangstr/hg38/genomewide/hg38_ver17.bed.gz.

The file str_info_file contains the per-locus stutter parameters obtained by training the stutter model on 19 samples using a modified version of HipSTR v0.6.2 (https://github.com/mikmaksi/HipSTR) with non-default parameters --stutter-model-only (to skip genotyping), --chrom (to run separately for each chromosome), --min-reads 20, and --output-filters. mergeSTR[47] v3.0.3 was used to merge the VCF files of each sample into a unified VCF file for each population.

**HipSTR.** We used HipSTR[8] v0.6.2 with non-default parameter --max-reads 2000000 to perform joint autosomal STR genotyping separately for each population using the hg38 STR reference available at https://github.com/HipSTR-Tool/HipSTR-references/blob/master/human/hg38.hipstr_reference.bed.gz.

**adVNTR.** adVNTR[7] v1.4.0 was run on each sample separately with a custom reference TR set (https://cseweb.ucsd.edu/~mbakhtia/adVNTR/vntr_data_recommended_loci_hg38.zip). For adVNTR's reference set, a total of 10,264 loci were selected. We started with TRs detected by Tandem Repeats Finder[46] to identify an initial set of VNTRs located in coding, untranslated, or promoter regions. To identify VNTRs in coding exons and UTRs, we used RefSeq gene coordinates downloaded from UCSC Table Browser[48]. For VNTRs within promoter regions, we considered 500 bp upstream of the transcriptional start sites of genes as the promoter regions. A total of 13,081 VNTRs were identified, of which 10,262 VNTRs were within the size range for short-read genotyping. We subsequently added two VNTRs known to be linked to human disease[49]. mergeSTR v4.0.1 was used to merge the VCF files of each sample into a single VCF file for each population.

**ExpansionHunter.** ExpansionHunter[12] v5.0.0 was run separately on each sample using a variant catalog of polymorphic STRs (https://github.com/Illumina/RepeatCatalogs/blob/release-v0.1.x/polymorphic_STR/hg38/polymorphic_STR.json). mergeSTR v4.0.1 was used to merge the VCF files of each sample into a unified VCF file for each population.

## Filtering initial TR genotypes
Prior to EnsembleTR calling, all population-level VCF files from each tool were leniently filtered with dumpSTR[47] v3.0.3 with the following options: --min-locus-callrate 0.75 (to remove TRs with low call rate), --min-locus-hwep 0.000001 (to remove TRs whose genotypes do not follow Hardy-Weinberg Equilibrium), and --filter-regions hg38_seg-dup.sorted.bed.gz --filter-regions-names SEGDUP (to remove TRs overlapping segmental duplications obtained from the UCSC Genome Browser[50]). For GangSTR, we additionally used options --gangstr-filter-spanbound-only and --gangstr-filter-badCI to remove low quality calls. For AdVNTR calls, we additionally devised a motif complexity filter, which discards TRs where the consensus motif has imperfect internal repeat structures. For example, a VNTR with consensus motif TTTTTCTT may actually be capturing a homopolymer repeat resulting in incorrect VNTR calls.

For this filter, we iteratively mask out characters in the motif and compute the Hamming distance between the masked motif against a sliding window of itself, while allowing the masked character to match any other character in the motif. Let $M$ be a motif of length $n$, we first compute $M'$ by masking $k$ characters. For a pre-selected $k$ and $i \in [0, \binom{n}{k} - 1]$ (using 0-based string indexing), $M'_{i,k}$ is defined as a string matching the original motif except characters that are in the $i$'th subset of length $k$ of the motif are masked.

For each $M'_{i,k}$, we concatenate it to itself to form $M''_{i,k}$ to identify any possible internal repetitions. The motif score is then computed as follows:

$$\text{Score}(M) = \max_{i \in (0, \binom{n}{k}-1), j \in (1, n-1)} \text{dist}(M'_{i,k}, M''_{i,k}[j : j + n - 1]) \tag{1}$$

where dist($x$,$y$) gives the Hamming distance between two strings $x$ and $y$ with equal length. The value $k$ is determined based on the motif length $n$: $k = \lceil \frac{n}{10} \rceil$ if $n \leq 40$, else 1.

Finally, Score ($M$) is normalized by the motif length $n$, to ensure values are between [0,1], with values closer to 1 indicating the presence of an internal repeat structure. We filtered all VNTRs with motif score > 0.8.

## Merging all populations
Finally, filtered population-level VCF files from each tool were merged using mergeSTR v4.0.1 to generate a single VCF file containing all samples. HipSTR in some cases adjusts the coordinates of an STR region to encompass polymorphic flanking regions around the repeat. In some cases, this can lead to the same STR having slightly different genomic coordinates in the VCF output for different populations. Thus, merging HipSTR VCF files across populations required specific modifications in mergeSTR code. When mergeSTR tries to merge records from different populations, three scenarios can happen: 1) A TR has the same starting position and reference allele sequence in populations A and B, in which case mergeSTR correctly merges them into one record. 2) A TR has different starting positions in populations A and B, in which case mergeSTR writes two distinct records for repeat X in the output. 2) A TR has the same starting position in populations A and B, but the end coordinate and reference allele sequence is different in these populations, in which case mergeSTR will skip the locus due to the inconsistency in the reference allele. To address this issue, we first modified mergeSTR to write the loci with different reference alleles and identical starting position as multiple records. The motified version of mergeSTR is available at (https://github.com/gymreklab/TRTools/tree/conf_ref). Then, a python script (https://github.com/gymreklab/1000Genomes-STRs/blob/main/Hipstr_correction.py) was used to correct the output VCF file of mergeSTR. First, all the records

with the same repeat ID are collected, then the largest overlapping region among all reference alleles is identified and all alleles are trimmed accordingly. If an allele sequence is empty after trimming, all genotypes with that allele were considered as no call. Genotypes are updated based on the new list of alleles and a corrected merged record is written in the output VCF file.

Finally, for GangSTR calls, after merging samples from all populations, we identified repeats with overlapping coordinates and among them, we only kept the first one.

**Ensemble genotyping**
EnsembleTR takes VCF files from multiple TR genotypers as input and outputs a merged consensus callset. The specific steps of EnsembleTR are described below.

**Identifying overlapping TRs between callsets.** EnsembleTR starts by finding the mutual samples across all callers. Then EnsembleTR walks through the list of TR loci (records) called by each method in sorted order to identify sets of mergeable calls. Records are deemed mergeable if they have overlapping coordinates and identical repeat unit sequences. EnsembleTR allows at most one record from each caller in each mergeable set.

**Matching alleles.** Mergeable sets may contain records from multiple callers, each of which might have genotyped a locus using slightly different representations of the possible alleles (see Fig. 1a for an example). To overcome this issue, EnsembleTR forms an internal representation of the consensus set of alleles such that alleles will be directly comparable across methods. It first extends all alleles to the maximum region spanned by all records. In this way, all alleles from different callers will start and end at the same position on the genome. It then extends the original alleles to span this entire region by prepending or appending flanking sequences extracted from the reference genome.

After identifying mergeable alleles, a representative sequence is determined for each allele set. If the mergeable set contains a HipSTR record, EnsembleTR uses the HipSTR allele sequences and discards alleles from other methods with the same length as the HipSTR alleles. This is done because HipSTR is the only method of the four used which reports the actual allele nucleotide sequence rather than only copy numbers. If there are two HipSTR alleles with the same length but different sequences in the allele set, both allele sequences are stored, and the original allele called by HipSTR for each sample is retrieved. In the case that an allele set contains two different HipSTR alleles, but a sample does not have a HipSTR call at that locus, we choose the most common allele of that length output by HipSTR. If a HipSTR record is not in the mergeable set, allele sequences are retrieved from one of the available callers randomly.

**Ensemble calling.** For each sample at each locus in a mergeable set, EnsembleTR matches calls from each method to the consensus alleles determined in the previous step. It then determines a consensus genotype by choosing the diploid genotype with the highest score as defined below. In case of ties, it gives priority to callers with the order of HipSTR, GangSTR, ExpansionHunter, and adVNTR. Let $S_g$ be the score for a diploid genotype $g$. $S_g$ is computed as

$$S_g = \frac{\sum_{m \in \mathbf{M}} Q_{g,m}}{\sum_{g' \in \mathbf{G}} \sum_{m \in \mathbf{M}} Q_{g',m}} * \max_{m \in \mathbf{M}} Q_{g,m} \tag{2}$$

Where $\mathbf{M}$ is the set of methods considered, $\mathbf{G}$ is the set of possible diploid genotypes (pairs of consensus alleles), and $Q_{g,m}$ is the quality score for method $m$ for genotype $g$. If the genotype returned by method $m$ is not equal to $g$, then $Q_{g,m}$ is set to 0. Otherwise, $Q_{g,m}$ is set to a quality score specific to each method. For HipSTR, AdVNTR, and

GangSTR the quality score is obtained from the Q score of the original VCF file, which ranges from 0 to 1. For ExpansionHunter, we defined a quality score based on the copy numbers and confidence intervals of each allele in the called genotype. Each allele's score is calculated by the formula:

$$\frac{1}{\exp(4 * \frac{CI}{CN})} \tag{3}$$

where $CN$ is the copy number and $CI$ is the length of the confidence interval. Then a final score for the ExpansionHunter genotype is calculated as a weighted average between two alleles' scores. A coefficient of 0.8 is used for the lower score and 0.2 for the higher one to give prominence to the low-quality genotype. We tried coefficients other than (0.8, 0.2) in score definition and compared their performance in terms of alignment with the Mendelian Inheritance rates in ExpansionHunter calls. While all settings of score coefficients are effective in capturing the true quality of calls according to MI error rates, differences in their performance are negligible. EnsembleTR outputs a new VCF file with the final genotypes with the highest score, along with the score $S_g$ for each call.

**Experimental validation of TR genotypes**
For each candidate TR, we obtained primers to amplify the TR and surrounding region (Supplementary Data 3). A universal M13(−21) sequence (5′-TGTAAAACGACGGCCAGT-3′) was appended to each forward primer. We then amplified each TR using a three-primer reaction previously described[51] consisting of the forward primer with the M13(−21) sequence, the reverse primer, and a third primer consisting of the M13(−21) sequence labeled with a fluorophore.

The forward (with M13(−21) sequence) and reverse primers for each TR were purchased through IDT. The labeled M13 primers were obtained through ThermoFisher (#450007) with fluorescent labels added to the 5′ ends (either FAM, VIC, NED, or PET). TRs were amplified using the forward and reverse primers plus an M13 primer with one of the four fluorophores with GoTaq polymerase (Promega #PRM7123) using PCR program: 94 °C for 5 min, followed by 30 cycles of 94 °C for 30 s, 58 °C for 45 s, 72 °C for 45 s, followed by 8 cycles of 94 °C for 30 s, 53 °C for 45 s, 72 °C for 45 s, followed by 72 °C for 30 min.

For several loci which were difficult to amplify using the above conditions, we used separate PCR conditions. Full experimental details for these loci are provided in Supplementary Methods. For HTT, C9orf72, and FMR1 we used available kits from Asuragen for genotyping (HTT: AmplideX® PCR/CE HTT Kit[52], C9orf72: AmplideX® PCR/CE C9orf72 Kit[53], FMR1: AmplideX® PCR/CE FMR1 Kit[54]).

Fragment analysis of PCR products was performed on a ThermoFisher SeqStudio instrument using the GSLIZ1200 ladder, G5 (DS-33) dye set, and long fragment analysis options. Raw PCR product sizes are given in Supplementary Data 4. Product sizes were converted to allele lengths using a binning process described in the Supplementary Methods.

Asuragen results are reported in Supplementary Data 5. To make Asuragen results (reported in total repeat copy number) comparable to WGS calls (reported as the number of repeat units relative to hg38), we applied an offset of −3 and −19 for C9orf72 and HTT genotypes, respectively. While AmplideX® PCR/CE FMR1 Kit results are also reported for FMR1, we did not include that locus in our validation analysis since our WGS calls include only autosomal loci.

For HTT only, we performed a sequence-specific analysis to compare WGS and experimentally validated genotypes. The repeat region in HTT consists of the sequence (CAG)nCAACAGCCGCCA(CCG)n. While EnsembleTR identifies changes in either the CAG or CCG repeat, the AmplideX® PCR/CE HTT Kit specifically analyzes only the CAG repeat. Therefore, we extracted the total number of CAG repeats, rather than the entire repeat length, from EnsembleTR before

performing comparisons. Notably, the SCA1 locus consists of an imperfect repeat. While EnsembleTR, HipSTR, and the capillary electrophoresis calls measure the total change in repeat length, GangSTR considers only the longest perfect repeat stretch, which likely accounts for the discrepancy with GangSTR calls at this locus.

## Published allele frequencies for forensics and disease-associated TRs

Population-specific allele frequencies for the CODIS forensics TRs in EUR, AMR, AFR, and EAS populations were obtained from NIST STRBase (https://strbase.nist.gov/, https://strbase.nist.gov/1036-Revised-Allele-Freqs-PopStats-July-19-2017.xlsx). SAS allele frequencies were obtained from literature sources[55,56]. Control allele frequencies for disease-associated TRs were obtained from various sources: (1) HTT: Validated repeat lengths were previously obtained[13] from Huntington's Disease patients (dbGaP accession "phs000371.v2.p1"). We used non-expanded alleles from table pht002988.v1.p1.c1 to estimate control allele frequencies in European samples. For other populations, allele frequencies were extracted from Masuda, et al. [57] (EAS), Baine, et al. (AFR and H3Africa)[58], Saleem, et al. [59] (SAS), and Paradisi, et al. [60] (AMR); (2) DMPK: Allele frequencies were obtained from Ambrose, et al. [61] (EAS), Acton, et al. [62] (AFR), and Magana, et al. [63] (AMR); (3) PPP2R2B: Allele frequencies for Europeans were obtained from Majounie, et al. [64] Similarity between published allele frequencies and those obtained from EnsembleTR was measured using Jensen-Shannon divergence, implemented in the python scipy library[65] v1.5.2. To obtain a null value for the divergence, for each locus/population pair, we first calculated the average Jensen-Shannon divergence between its EnsembleTR frequency and the published frequency for all other locus/population pairs. We then calculated the mean over all pairs.

## Mendelian inheritance analysis

We analyzed 602 trios available in 1000GP. For each trio, we only assess the Mendelian Inheritance if 1) calls were available for all three samples and 2) at least one of the samples is not homozygous for the reference allele. The score assigned to each trio is the minimum score reported by EnsembleTR among all samples in the trio. In all analyses except the TR expansion analysis, TRs with Mendelian error rates >5% were filtered, leaving 1,443,686 total TR loci.

## Characterization of population-specific TR variation

We performed principal component analysis on a matrix $\mathbf{X}_{n,m}$ where $n$ is the number of TR loci and $m$ is the number of samples. Each cell $c_{i,j}$ of $\mathbf{X}$ denotes the sum of allele lengths for a diploid call of the $j$ th sample at the $i$ th locus. In the case of a no call, $c_{i,j}$ is set to NaN. Due to the large size of $\mathbf{X}$, we used a memory-efficient method, Incremental Principal Component Analysis (IPCA), implemented in the python scikit-learn library[66] v1.0.2.

## Inspecting TRs not matching hg38

To validate the 196 TRs for which the majority of alleles ( > 95%) differ by more than 2 repeat units from the hg38 reference (Supplementary Data 10), we examined the length of those TRs separately in both the T2T[23] reference and PacBio HiFi reads for sample HG00438 obtained from the Human Pangenome Reference Consortium[67] (HPRC) (https://s3-us-west-2.amazonaws.com/human-pangenomics/index.html?prefix=working/HPRC/). For the PacBio HiFi dataset, we used the Integrative Genomics Viewer[68] to manually inspect reads aligning to each TR. To compare to the T2T reference v1.1 (https://s3-us-west-2.amazonaws.com/human-pangenomics/T2T/CHM13/assemblies/chm13.draft_v1.1.fasta.gz), hg38 coordinates of the 196 TRs were converted to T2T v1.0 first and then converted to v1.1 using the UCSC liftOver[50] utility with the corresponding chain files at http://t2t.gi.ucsc.edu/chm13/hub/t2t-chm13-v1.0/hg38Lastz/hg38.t2t.chm13-

v1.0.over.chain.gz and https://s3-us-west-2.amazonaws.com/human-pangenomics/T2T/CHM13/assemblies/changes/v1.0_to_v1.1/v1.0_to_v1.1_rdna_merged.chain. For TRs that failed to convert due to partial deletion, we added additional flanking sequences (up to 1000 bp) to the start and end coordinates and reattempted liftOver, which resulted in successful conversion of 195/196 TRs to T2T v1.1. One TR failed due to deletion in the T2T v1.1 reference. For each TR, We used samtools[69] v1.5 to extract its sequence with flanking regions from both the T2T v1.1 and hg38 references and compared the two sequences using BLASTN[70] v2.13.0 + . We used a similar method to compare the TR lengths between the T2T v1.1 reference and the major alleles called in EnsembleTR by replacing the hg38 reference with the EnsembleTR major allele at each TR.

## Detecting population-specific expansions

For each TR with repeat unit length >1 bp, we defined an expanded allele to be any allele with copy number greater than $Q3 + 3*IQR$, where Q3 is the third quartile, and IQR is the difference between the third and first quartile. We calculated the frequency of expansions in African (1000GP AFR and H3Africa) and Non-African populations (all other 1000GP super-populations). We defined TRs with population-specific expansions as those for which 1) the expansion threshold copy number $(Q3 + 3*IQR)$ is greater than 10, 2) the frequency of expansions is greater than 0.01 in at least one population and, 3) the expansion frequency in one population is at least 10 times larger than the other population. Gene annotations for repeat expansions were based on Ensembl version 108 available at https://ftp.ensembl.org/pub/release-108/gtf/homo_sapiens/Homo_sapiens.GRCh38.108.gtf.gz.

To support these results, we applied two additional TR genotypers (STRetch[24] and ExpansionHunter Denovo[11]) to identify expansions in the H3Africa cohort. STRetch v.0.4.0 takes as input a reference genome with decoy STR contigs of length 2000bp, a CRAM or BAM file and a bed file with genome locations of TRs. A STRetch STR catalog was generated for GRCh38 and the recommended pipeline for WGS was run for each sample. We ran STRetch twice, once using 143 control samples provided and once without any controls and using all samples in one batch. However, due to the distinct nature of the sequencing data for control samples in STRetch, which were PCR-free unlike the H3Africa samples, we consider the results from the second analysis to be more reliable and therefore use them for downstream analysis. STRetch results for each sample were then merged into a single file and filtered using the criteria p_adj <0.05, locuscoverage >=3 and outlier Z-score >= 8. The filtered loci were annotated using the OrganismDbi[71] R package v1.40.0.

ExpansionHunter Denovo v.0.9.0 was used to generate genome-wide STR profiles for each sample with default parameters of --min-unit-len 2, --max-unit-len 20, --min-anchor-mapq 50, and --max-irr-mapq 40 in order to restrict the search of motif lengths of up to 20 bp. Unlike STRetch, ExpansionHunter Denovo does not require prior knowledge of the location of repeats in the genome. A manifest file was synthesized where each sample was labeled as a case and STR profiles were merged to allow comparisons among samples after read depth normalization. Outlier locus and motif analyses were performed using scripts available in the ExpansionHunter Denovo package and the output was ranked using Z scores. Annotation of locus-based analysis results was done using ANNOVAR[72]. To find the overlap between our set of expansions and expansions found by ExpansionHunter Denovo and STRetch, for each repeat expansion $r$ in our list, we checked if there is any expanded repeat reported by either ExpansionHunter Denovo or STRetch that meets two conditions: 1) it is located in the surrounding ±1000 bp window of $r$, 2) the repeat unit sequence of both expansions are the same.

To validate the candidate repeat expansion in an intron of *CA10*, we designed primers to amplify the TR and surrounding region using the three-primer reaction described above. consisting of the forward

primer with the M13(-21) sequence (5′-TGTAAAACGACGGCCAGTTG GCTCCAAGTAGCACATCTT-3′), the reverse primer (5′-TGCAACTAG CGGTGACCTTA-3′), and a third primer consisting of the M13(-21) sequence labeled with a fluorophore. Primers were purchased through IDT. The labeled M13 primers were obtained through ThermoFisher (#450007) with FAM fluorescent labels added to the 5′ ends. The locus was amplified with GoTaq polymerase (Promega #PRM7123) using the PCR program: 94 °C for 5 min, followed by 30 cycles of 94 °C for 30 s, 59 °C for 45 s, 72 °C for 45 s, followed by 8 cycles of 94 °C for 30 s, 53 °C for 45 s, 72 °C for 45 s, followed by 72 °C for 30 min. We tested on 4 samples from 1000GP (HG01119, NA20847, NA19434, NA12878). Fragment analysis of PCR products was performed on a ThermoFisher SeqStudio instrument using the GSLIZ1200 ladder, G5 (DS-33) dye set, and long fragment analysis options. We used the reference allele AAGCAGCAGCAGCAGCAGCAGCAGCAGCAGCAGCAGCAGCAGCAGC AGCAGCAGCAGCAGCAGCAG (22 repeats) to set up bins for analysis using the Genemapper software.

## Visualizing repeat composition with TRviz
We used TRviz v1.0.1[26], a python library to visualize TR alleles, for a subset of 27 1000GP samples.

## Finding sequence determinants of TR heterozygosity
We summarized the variability of each TR using heterozygosity, computed as $1 - \sum_{i=1}^{n} p_i^2$, where $p_i$ is the frequency of allele $i$ and $n$ is the total number of alleles. Heterozygosity was computed using the statSTR utility from TRTools[47] v4.2.1 with flags --het --vcftype hipstr.

For classifying dinucleotide TRs as stable vs. polymorphic, we used pure (no sequence imperfections) dinucleotide STRs with lengths of 12-17 bp based on the HipSTR STR reference set that were genotyped in at least 3000 samples between 1000GP and H3 Africa. The 82 perfect 'GC' motif STRs were omitted to prevent overfitting and due to the low number of those TRs.

HOMER[30] v4.11.1 was run twice for each TR repeat unit type (AC/GT, AT, and AG/CT), alternating having the variable or stable STRs as the foreground set and the other as the background. For each sequence, we input either the forward or reverse complement sequence such that the TR repeat unit matched the canonicalized repeat unit sequence (AC for AC/GT repeats, AT for AT repeats, and AG for AG/CT repeats). The input sequences for each group were the 64 bp flanks on either side of the TRs. The sequences before and after each STR were separate examples but part of the same variable or stable set. HOMER was run to find motifs 4-12 bases long without GC correction with the command findMotifs.pl <targetSequences.fa> fasta <output directory > -fasta <background.fa > -len 4,5,6,7,8,9,10,11,12 -noweight.

Our neural network model consists of a 1-D CNN with inception blocks[73] implemented with Pytorch. Instead of applying convolutional kernels of a single width, inception blocks in parallel apply filters of multiple widths in addition to a pooling and single-width convolution. Our best performing model used six inception blocks with kernel sizes 5, 9, and 15 followed by global average pooling and a single linear layer. The data was split 70:15:15 into train, validation, and test splits and reverse complements were added to the same split as the forward strand sample.

To blind the model from STR length and focus on nearby sequence information, the model input is the 58 bp flanking the STR and the 6 directly adjacent bases that make up the STR from both flanks. The model input is then these two flank sequences concatenated together into a single sequence. The input was represented using one-hot encoding, so a zero matrix was used as a baseline for generating attribution scores with Integrated Gradients[74]. We found that using a global average pooling layer instead of a global max pooling layer led to more informative attribution scores.

## eQTL analysis in the GEUVADIS cohort
We obtained gene-level reads per kilobase of transcript per million mapped reads (RPKM) values for 452 unrelated individuals generated from lymphoblastoid cell lines by the GEUVADIS project[31] (https://www.internationalgenome.org/data-portal/data-collection/geuvadis). Duplicated samples were removed by arbitrarily keeping the first dataset for each. Genes with RPKM above 0.1 in more than 10 samples were kept for downstream analysis. Expression values for remaining genes were quantile-normalized on sample level followed by quantile-normalization to a standard normal distribution separately for each gene. Genes overlapping segmental duplications were removed, and analysis was restricted to protein-coding genes based on GENCODE v12 annotation. Gene coordinates were adjusted from hg19 to hg38 using the liftOver available from the UCSC Genome Browser[50]. After filtering, 12,607 genes remained for analysis.

To control for population structure, we obtained publicly available genotype data on 2318 unrelated individuals from the 1000GP genotyped with Omni 2.5 SNP genotyping arrays (http://ftp.1000genomes.ebi.ac.uk/vol1/ftp/release/20130502/supporting/hd_genotype_chip/). We removed all indels, multiallelic SNPs, and SNPs with a minor allele frequency of less than 5%. We then used plink[75] v.1.90b3.44 to subset these remaining SNPs to a set of SNPs in approximate linkage equilibrium with the command --indep 50 5 2. We excluded any remaining SNPs with a missingness rate of 5% or greater. We lastly ran principal component analysis using smartpca[76,77] v.13050 with default parameters.

Association tests were performed separately on the African and European populations. For each TR, we tested for association with each gene within 100 kb. We performed a linear regression for each test between the TR dosage (the sum of allele lengths relative to the hg38 reference genome) and normalized gene expression. We included the top 10 genotype principal components as computed above, 44 PEER factors[78], and sex as covariates. The number of PEER factors was chosen based on the recommended 1/10 of the sample size. PEER factors were calculated using PEER v1.0 based on the normalized gene expression data of all 452 samples. In cases where fewer than 50 samples had non-missing TR genotypes, the TR was removed from analysis in that population.

To identify individual significant eTRs, we obtained adjusted p-values using the Benjamini-Hochberg approach for controlling the false discovery rate[79] applied to P-values for all TR-gene pairs separately in Europeans and Africans. To identify gene-level significant eTRs, we followed the steps in our previous eTR analyses[16,32]. For each gene, we determined the TR association with the strongest P-value. This P-value was adjusted using a Bonferroni correction for the number of TRs tested per gene to give a P-value for observing a single eTR association for each gene. We then used the list of adjusted P-values (one per gene) as input to the Benjamini-Hochberg method to obtain a q-value for the best eTR for each gene.

eTR summary statistics based on the GTEx dataset used for effect size comparisons were obtained from Supplementary Dataset 2 of Fotsing et al.[16]. TR coordinates were lifted over to hg38 for comparison. Because coordinates can vary slightly between callsets, we identified TRs as overlapping if their coordinates were within 20 bp.

## Phasing and imputation
Phased SNPs for 1000GP samples were downloaded from the 1000GP FTP server (http://ftp.1000genomes.ebi.ac.uk/vol1/ftp/data_collections/1000G_2504_high_coverage/working/20201028_3202_phased). We used Beagle v4.0[80] to phase each TR separately. To produce a high-quality TR callset for phasing, we performed additional filtering to remove all calls with quality score below 0.9, TRs with call rate below 0.8, and TRs with average Mendelian Error rate > 5%. We manually included three pathogenic loci (HTT, SCA1, and SCA3) that did not meet the filtering criteria.

Our pipeline is based on our previously published framework[35] and takes the unphased TR and surrounding phased SNPs from a 50 kb window centered at the TR as input (--gt). We set the --usephase parameter to True to allow Beagle to use the phase information of provided phased SNPs to phase the target TR. This step outputs a phased VCF file containing both SNPs and the target TR. We apply a custom script to ensure the phase order matches the original SNP input. We then extract and concatenate phased TR genotypes from each locus and combine them with the original phased SNPs into a single phased VCF file. In order to comply with Beagle's requirement of a maximum of 126 alleles per record, manual modifications were made to the VCF entries for HTT and SCA1. The alteration involved unifying alleles with the same length but different sequences, and subsequently updating the corresponding genotypes to meet the specified requirement.

We then used Beagle v5.4[81] to perform a Leave-One-Out analysis to assess concordance. This analysis was restricted to Chromosome 21 due to the high computational burden. For each sample $S$, phased SNPs +TRs for all samples except $S$ are given to Beagle as --ref and phased SNPs for sample $S$ are given to Beagle as reference panel (--gt). Beagle will use these inputs to impute the missing TRs for $S$. After performing imputation for $n = 100$ randomly chosen samples from each population (excluding trio samples), concordance for each locus is computed as follows: For each sample $S_i, i \in \{1..n\}$, let $\mathbf{x}_{ij}$ be the EnsembleTR genotype and $\mathbf{y}_{ij}$ be the imputed TR genotype for sample $S_i$ at the $j$ th locus. Each genotype for a diploid sample contains two alleles, therefore we will define $\mathbf{x}_{ij} = (x_{ij1}, x_{ij2})$ and $\mathbf{y}_{ij} = (y_{ij1}, y_{ij2})$. Then concordance $c_{ij}$ for $S_i$ at the $j$ th locus is computed as: 1 if both genotypes match: $\text{sorted}(x_{ij1}, x_{ij2}) == \text{sorted}(y_{ij1}, y_{ij2})$; 0 if neither imputed allele matched an EnsembleTR allele; else 0.5 if one but not both imputed alleles matched the EnsembleTR alleles. Total concordance for the $j$ th locus $c_j$ is computed by averaging over concordance values for each sample $c_j = \frac{1}{n} \sum_i c_{ij}$.

To compute LD between each TR and a nearby SNP, we calculated the squared Pearson correlation coefficient between the SNP and TR genotype vectors, where each vector has 2n elements, n is the number of samples and each sample has two alleles. We used the phased and imputed TR genotypes for this analysis and we selected the SNP with the highest r² as the candidate tag SNP for each target TR.

To evaluate imputation of trait-associated TRs (Supplementary Data 17), we obtained coordinates of 95 blood-trait associated loci from Supplementary Table 4 of Margoliash, et al. [36]. TR coordinates were converted from hg19 to hg38 using the UCSC liftOver tool[50].

## Reporting summary
Further information on research design is available in the Nature Portfolio Reporting Summary linked to this article.

## Data availability
TR genotypes, the phased TR-SNP reference panel, and population-specific summary statistics generated in this study are available from the EnsembleTR GitHub webpage [https://github.com/gymrek-lab/EnsembleTR]. Summary statistics are also made available in browsable format at WebSTR [https://webstr.ucsd.edu]. Summary statistics of all tested TR-gene pairs in African and European samples are deposited in the Figshare database [https://figshare.com/articles/dataset/1000GenomesH3Africa_SuppData16_zip/24164367]. The WGS datasets for the 1000GP samples used in this study are available from the European Nucleotide Archive under accessions PRJEB31736 (unrelated samples) and PRJEB36890 (related samples). The H3Africa WGS datasets used in this study are available in the European Genome-Phenome Archive under accession EGAS00001002976. Geuvadis datasets used in this study are available from the 1000GP website [https://www.internationalgenome.org/data-portal/data-collection/geuvadis]. Analyses are based on the GRCh38 reference genome [https://storage.googleapis.com/genomics-public-data/resources/broad/hg38/v0/Homo_sapiens_assembly38.fasta].

## Code availability
EnsembleTR is available on GitHub: https://github.com/gymrek-lab/EnsembleTR[82].

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

## Acknowledgements

This work was partially funded by NIH/NHGRI grants 1R01HG010149 (M.G. and V.B.), R01HG010885 (M.G.), 1RM1HG011558 (M.G.), 5U24HG006941 (Y.A. and E.A.), 5U2RTW010679 (E.A.), 1U2RTW010672-01 for (D.J, I.L, F.K and H.J) and 1U2CEB032224-01 (D.J). B.O. and J.A. were partially funded by the World Bank ACE Impact funding for CApIC-ACE. V.B., J.P., and S.J. were supported in part by grants GM114362, HG010149, and RM1HG011558. The authors acknowledge Brian Haynes and Sarah Statt for providing Asuragen kits and results and Rahel Wachs for help with illustrations. The authors also acknowledge support from the H3Africa genome analysis working group, as well as TrypanoGEN, CAfGEN and Baylor for data from the H3Africa consortium.

## Author contributions

H.Z.J. developed EnsembleTR, led analyses, and wrote the manuscript. Y.L. performed genotyping in the H3Africa cohort and led analysis of eTRs. R.D. performed analysis of sequence determinants of TR variability. N.Mo. developed the initial version of EnsembleTR and performed genotyping in the 1000GP cohort. N.Ma performed experimental validation of TR genotypes and helped with analysis of population-wide patterns of TR variation. I.L. performed analysis of repeat expansions in the H3Africa cohort. Y.A. helped with EnsembleTR genotyping in the H3Africa cohort. M.M. developed an improved GangSTR reference panel. B.H. helped with analysis of experimentally validated genotypes in 1000GP pedigrees. E.D. and Y.Q. helped with ExpansionHunter genotyping. F.E.K. helped perform comparisons between EnsembleTR and existing databases of genetic variation. H.J. helped with the principal component analysis. B.O. and J.A. helped with analysis of population-wide patterns of TR variation. M.B., J.P., and S.J. helped perform adVNTR genotyping in the 1000GP data. J.P. helped perform adVNTR genotyping in the H3Africa data and visualization of alleles at repeat expansion loci with TRviz. D.J. supervised H3Africa analyses and identification of novel repeat expansions in African genomes. E.A. supervised H3Africa analyses and contributed to the manuscript. V.B. supervised analysis of VNTRs. M.G. supervised development of EnsembleTR, experimental validation, and analyses and wrote the manuscript.

## Competing interests

V.B. is a co-founder, consultant, SAB member and has an equity interest in Boundless Bio, inc. and Abterra, Inc. The terms of this arrangement have been reviewed and approved by the University of California, San Diego in accordance with its conflict of interest policies. E.D. and Y.Q. are employees of Illumina, Inc., a public company that develops and markets systems for genetic analysis. The remaining authors declare no competing interests.
