## [Peer Review File · Nature Communications]

A deep population reference panel of tandem repeat variationREVIEWER COMMENTS

Reviewer #1 (Remarks to the Author):

Ziaei Jam and colleagues explore tandem repeat variation between populations in ~3500 samples from 1000 Genomes and H3Africa using a new method, EnsembleTR, which they developed to combine TR calls from four different short read methods. They identify population-specific repeat expansions and flanking sequencing features associated with heterozygosity of dinucleotide repeats. They also expand on previous work, identifying TRs associated with expression and creating a panel for imputing polymorphic STRs.

I tested EnsembleTR and found the software very easy to install and use. The TR calls and the STR imputation reference panel provided on the GitHub page are useful resources.

A major limitation of this study is the difference in the TR catalogs used with each TR genotyping method. More than half of the TR loci are genotyped by only a single method and only ~1% by three or more methods. Using a consistent catalog between all methods would significantly improve this work.

The ensemble approach developed in this study is potentially very valuable, however the evaluation of EnsembleTR is limited by inconsistency in catalogs between methods. The observed Mendelian inheritance rate is generally higher when there are more input methods, however it is unclear whether this is due to differences in accuracy between methods, the improvement provided by EnsembleTR, or other factors such as differences in the composition of TR loci genotyped by each method. Comparing the performance of EnsembleTR applied to different combinations of input methods versus individual methods alone, across the same set of input TR loci for all methods would help to demonstrate the performance of EnsembleTR and which methods provide the most accurate TR genotypes.

The ensemble approach is evaluated by validating a small subset of samples and TRs. It would be useful to compare the performance of EnsembleTR and each individual method (as suggested above). This data is available in supplemental tables and figures but is not mentioned in the main text.

I would also be interested to see a summary of the number/percentage of TR loci where genotype calls differed between individual methods, for which EnsembleTR could improve genotyping accuracy.

The first paragraph on the Discussion states: "Ensemble genotyping results in high-quality genotypes at more than 1.7 million TR loci, far more TRs than are successfully genotyped by any single method." I disagree with this statement given that more than 50% of TRs were only based on a single method, thus ensemble genotyping did not add any quality improvement to most loci. The number of TRs genotyped by a single method is also based on the user choice of catalog, which can be increased if desired.

In "A genome-wide catalog of TR variation" subsection of the Results: "We examined population-specific allele frequencies at well-characterized TRs...and found that EnsembleTR results recapitulated published results for these loci"

Describe how the consistency between observed results and previously published results was evaluated. Was this determined by a particular metric or by eye based on the distributions?

In the "Detecting TRs associated with gene expression" subsection of the Results: "Effect sizes computed for overlapping sets of TR-gene pairs across studies were significantly correlated in all tissues...but were most strongly correlated with Cultured Fibroblasts."

Is the data that this result is based on included in the paper? I do not see any details about correlations for different tissues.

Nearly all figures in the main body of the manuscript contain small text in places which is difficult to read. Small text should be made larger, or figures reorganized to be easier to read.

Figure 2c appears to have a non-linear (log?) scale on the y-axis. More y-axis labels are needed to determine the scale. Figures 2d and 2e might benefit from having a similar type of non-linear scale as it is nearly impossible to resolve the detail of the distribution for large repeat numbers.

Figure 3g could contain fewer numbers, making it easier to read. Presenting p-values and log p-values is unnecessary, and values do not need to be given to four significant figures.

Supplemental Table 9 has a column named "EH_supported". Should this be EHDN rather than EH, as the Methods refer to ExpansionHunter Denovo, not ExpansionHunter?

The Methods describe the construction process for the catalogs used with HipSTR, GangSTR and adVNTR. Are these catalogs available on the respective webpage for each method? If so, list the version and/or URL.

The "Detecting population-specific expansions subsection" of the Methods that describes STRetch is confusing. It states that each sample was compared to the controls provided with STRetch and describes ten WGS samples. The STRetch paper includes 10 samples with known pathogenic expansions, which match the description provided here, but also a larger number (~100) of controls. STRetch can also be run without controls, using all samples in a batch instead, which would likely give better results than a set of only 10 controls.

In the Filtering initial TR genotypes subsection of the Methods: "some regions are called by both adVNTR and an STR caller...tend to have lower Mendelian error rates in adVNTR". This is likely a typo ("lower"  "higher")

Reviewer #2 (Remarks to the Author):

The authors did a major effort to characterize TR in the human genome at population level. This represents the largest and most comprehensive catalog of TR sizes ever published, well beyond other efforts in this direction and it will be of great use to many researchers. Comments and suggestions:

- limitations of the approach, related to short reads should be clearly stated

- They validated a subset of the database calls (something like 48 sites on each of 48 samples) by capillary electrophoresis and saw largely good results, but the discrepancies were clustered on a CG-pure locus (C9ORF72) and heterozygous loci where they saw dropout of one allele. The authors argued that "the vast majority of TRs genotypes based on WGS are of comparable accuracy to those obtained by the experimental gold standard of fragment analysis", but it isn't really clear to me whether this is true. Perhaps the majority of the catalog are loci that are small, non-CG-pure, and infrequently heterozygous/polymorphic. It should be discussed what percentage of the TR catalog are gc-pure loci which may be more prone to incorrect calling. If this is a very small percentage, they should discuss whether that is a biological or a technical theme. And if it is not a very small percentage, the 'vast majority' statement should be toned down.

- It is important to better disambiguate 'TRs' from 'their catalog of TRs'. In particular, authors need to address the fact that lots of repetitive regions in the genome exist that are big and messy and are almost certainly systematically excluded from their catalog (e.g., anything over 1kb).

- Please state the ratio of homopolymers to all other length k-mers in the catalog. it is hard it is to tell their relative sizes.

- The main deliverable of this manuscript is a database of TR sizes, but its difficult to access. Authors should offer a download of the allele sizes (or at least a few percentiles of sizes for each allele) as a text file either as a supplementary dataset or through their website. But also, the website is still very much a work in progress, particularly for GRCh37, so at this point I don't feel like the database is really being delivered to readers in any effective way.

- It is unclear what result is returned in this catalog if a locus has a motif change that is not explicitly encoded in a tool like ExpansionHunter (like if there is a locus similar to RFC1 that hasn't been well-studied yet). They make a lot of arguments that HipSTR handles that, but only if it is below ~150bp, correct? What happens if the allele is larger?

- the explanation of the classifier performance in the 'sequence determinants of TR polymorphism' section is difficult to interpret. Due to the probable class imbalance, authors need to give quite a bit more detail here --actual accuracy, precision and recall values rather than just F1-scores, as well as area under the precision-recall curves would be nice.

- more generally, does this 4-mer observation explain a subset of sites well or a large number of sites weakly? I would appreciate some numbers of what percentage of their stable vs polymorphic loci exhibited this pattern. The quantification with spearman is nice, but doesn't really tell a lot.

- then utility of imputation TRs with nearby SNPs is somewhat overstated. As expected this is largely confined to stable TRs. The medically (GWAS) most interesting TRs are the unstable ones, especially the rare expansions. There is not a god reason to believe that nearby SNPs would capture this well (other than in founder haplotypes). Also given the meiotic instability and dynamism of expansions. This should be made more clear. I am not convinced (yet) that this is a true advance for GWAS studies.

Overview

We thank the reviewers for their careful reading of our manuscript and their insightful suggestions and comments which highlighted the utility of this resource and the fact that it is the “largest and most comprehensive catalog of TR sizes ever published”. We were encouraged by the positive feedback, and implemented the suggestions in our current manuscript version. We highlight here the major areas addressed in our revision.

1. More detailed comparisons of EnsembleTR vs. individual TR genotyping methods: Our initial manuscript focused on the lack of overlap between TRs considered by individual methods, and therefore highlighted the gain in the total number of TRs we could achieve using an ensemble approach. Based on reviewer comments, we have added new analyses to examine whether the actual call quality is improved at TRs that were called by more than one method. By examining TRs genotyped by more than one method, we found that EnsembleTR typically results in genotypes that more consistently follow Mendelian Inheritance compared to genotypes based on a single method. These analyses are described in more detail in our response to Reviewer #1 and summarized in new **Supplementary Tables 1-2**.

2. Improved data accessibility: Genotypes are now available in browsable format in our web resource, webstr.ucsd.edu (under hg38). We also now provide downloadable text files with population-specific per-locus summary statistics, in addition to updated VCF files with individual-level calls, on the EnsembleTR GitHub (<https://github.com/gymrek-lab/EnsembleTR>).

In the process of addressing comments, we produced an updated callset. The updated calls which include an expanded set of VNTRs (1,375 vs. 4,371) based on adjusting our filtering criteria for those to exclude “STR-like” repeats that were annotated as VNTRs. This new filtering strategy is described in **Methods**. All analyses and figures have been updated to use these new calls. We also note that the analysis of context sequence features contributing to heterozygosity of dinucleotide STRs previously was based on an outdated callset, but has been updated to use the most recent EnsembleTR genotypes.

Responses to additional comments: In-depth responses to each comment from reviewers are below. Reviewer comments are shown in blue and our responses are in black. Revised text in the main text is marked in track changes.

Reviewer 1 Comments for the Author...

Ziaei Jam and colleagues explore tandem repeat variation between populations in ~3500 samples from 1000 Genomes and H3Africa using a new method, EnsembleTR, which they developed to combine TR calls from four different short read methods. They identify population-specific repeat expansions and flanking sequencing features associated with heterozygosity of dinucleotide repeats. They also expand on previous work, identifying TRs associated with expression and creating a panel for imputing polymorphic STRs.

I tested EnsembleTR and found the software very easy to install and use. The TR calls and the STR imputation reference panel provided on the GitHub page are useful resources.

We thank the reviewer for these comments and are glad the resources were easy to use!

A major limitation of this study is the difference in the TR catalogs used with each TR genotyping method. More than half of the TR loci are genotyped by only a single method and only ~1% by three or more methods. Using a consistent catalog between all methods would significantly improve this work.

Due to the complexity involved in characterizing TRs, each TR genotyper is typically designed to quantify a subset of repeats with specific features. For instance, GangSTR is designed specifically for genotyping perfect repeats, where the sequence consists of a single motif. However, it can also be utilized to genotype repeats with motif lengths up to 20bp. On the other hand, HipSTR is capable of genotyping repeats with sequence imperfections, but it is limited to STRs where the motif length is at most 6bp. AdvNTR, on the other hand, is specifically designed for genotyping VNTRs, which often possess distinct structures compared to STRs and necessitate specific considerations. These variations in the design of these tools result in differences in the reference sets used by each one. Typically, each tool is published with the maximal reference set it is intended for. This lack of overlap was one of the major factors that motivated us to develop an ensemble approach. Therefore, we designed EnsembleTR to bring all of these TRs together in a single dataset with the aim of providing a comprehensive catalog of TRs. We now mention this in the **Introduction** and expand on these points in the **Discussion** section.

The ensemble approach developed in this study is potentially very valuable, however the evaluation of EnsembleTR is limited by inconsistency in catalogs between methods. The observed Mendelian inheritance rate is generally higher when there are more input methods, however it is unclear whether this is due to differences in accuracy between methods, the improvement provided by EnsembleTR, or other factors such as differences in the composition of TR loci genotyped by each method. Comparing the performance of EnsembleTR applied to different combinations of input methods versus individual methods alone, across the same set of input TR loci for all methods would help to demonstrate the performance of EnsembleTR and which methods provide the most accurate TR genotypes.

We agree in our initial manuscript we focused largely on the gain in quantity, rather than quality, of calls from Ensemble calling. We have added two analyses to determine what is driving the higher rates of Mendelian inheritance (MI) at loci called by multiple methods.

First, to explore whether the gain at these loci could be explained by differences in reference composition (e.g. if TRs included in references for multiple tools have properties that make them easier to genotype than TRs specific to a single tool), we performed MI analysis across TRs genotyped by each tool, stratified by how many tools included that TR in the reference (new **Supplementary Table 1**). These analyses show that TR composition does play a role: TRs included in reference panels of 3 tools in all cases tended to show higher MI even within a single method compared to TRs included by only a single tool.

Second, to determine whether EnsembleTR calling actually increases call quality beyond that which can be explained by TR composition, we examined MI at TRs on Chr1 genotyped by multiple tools (new **Supplementary Table 2**). We found that for all combinations of tools, EnsembleTR calling either exceeded MI for any individual tool or matched the quality of the best tool. Overall, our results show that EnsembleTR calling gains are due to multiple sources: some of this apparent gain at loci called by multiple tools is due to reference TR composition.

However, even at TRs called by multiple methods, EnsembleTR calls show higher MI than any individual input method.

The ensemble approach is evaluated by validating a small subset of samples and TRs. It would be useful to compare the performance of EnsembleTR and each individual method (as suggested above). This data is available in supplemental tables and figures but is not mentioned in the main text.

We added the concordance value for GangSTR and HipSTR vs. experimentally validated genotypes to the main text.

I would also be interested to see a summary of the number/percentage of TR loci where genotype calls differed between individual methods, for which EnsembleTR could improve genotyping accuracy.

We now report the percent of calls for each combination of methods that are discordant in **Supplementary Table 2**. In most cases, the methods are highly concordant. Still, as described above EnsembleTR results in improved MI rates compared to any individual caller alone.

We also found that in 72% of calls that were genotyped by multiple methods but where at least one method did not follow MI, EnsembleTR calling resulted in Mendelian consistency, compared to 65% obtained by a naïve approach of always choosing the HipSTR genotype. This result is now reported in the main text.

The first paragraph on the Discussion states: “Ensemble genotyping results in high-quality genotypes at more than 1.7 million TR loci, far more TRs than are successfully genotyped by any single method.”

I disagree with this statement given that more than 50% of TRs were only based on a single method, thus ensemble genotyping did not add any quality improvement to most loci. The number of TRs genotyped by a single method is also based on the user choice of catalog, which can be increased if desired.

We have revised this sentence to just state the number of loci genotyped “After filtering low quality calls, our ensemble approach genotyped more than 1.7 million TR loci.” and expand on the source of the gains in the quantity and quality of calls in the EnsembleTR results in a new paragraph added to the **Discussion**.

In “A genome-wide catalog of TR variation” subsection of the Results: “We examined population-specific allele frequencies at well-characterized TRs...and found that EnsembleTR results recapitulated published results for these loci”

Describe how the consistency between observed results and previously published results was evaluated. Was this determined by a particular metric or by eye based on the distributions?

Previously, this was based on visual inspection of allele length histograms. We have now provided a quantitative comparison based on the Jensen-Shannon divergence. The average divergence is reported in the text, and values for each locus in each population are annotated on the plots.

In the “Detecting TRs associated with gene expression” subsection of the Results: “Effect sizes computed for overlapping sets of TR-gene pairs across studies were significantly correlated in all tissues...but were most strongly correlated with Cultured Fibroblasts.”

Is the data that this result is based on included in the paper? I do not see any details about correlations for different tissues

We had computed these numbers but the actual values were not previously included. We have now added **Supplementary Table 16** which reports the correlation values for each tissue.

Nearly all figures in the main body of the manuscript contain small text in places which is difficult to read. Small text should be made larger, or figures reorganized to be easier to read.

We apologize for the small text. We have now revised all figures to ensure text is as large as possible, and have also provided separate high resolution pdf files.

Figure 2c appears to have a non-linear (log?) scale on the y-axis. More y-axis labels are needed to determine the scale. Figures 2d and 2e might benefit from having a similar type of non-linear scale as it is nearly impossible to resolve the detail of the distribution for large repeat numbers.

Fig. 2c is on a linear scale. We have added more y-axis ticks to make this more clear. For the histograms in **Fig. 2d-e**, we now combine all expansions to a single category to make it easier to read.

Figure 3g could contain fewer numbers, making it easier to read. Presenting p-values and log p-values is unnecessary, and values do not need to be given to four significant figures.

We have updated **Fig. 3g** to include only the necessary information (motif, p-value, % of targets, % of background). Previously this was a screenshot of Homer results, but has been manually redrawn in order to simplify it.

Supplemental Table 9 has a column named “EH_supported”. Should this be EHDN rather than EH, as the Methods refer to ExpansionHunter Denovo, not ExpansionHunter?

That is correct. This is now fixed.

The Methods describe the construction process for the catalogs used with HipSTR, GangSTR and adVNTR. Are these catalogs available on the respective webpage for each method? If so, list the version and/or URL.

We have added links to the reference sets for those tools in the **URLs** section.

The “Detecting population-specific expansions subsection” of the Methods that describes STRetch is confusing. It states that each sample was compared to the controls provided with STRetch and describes ten WGS samples. The STRetch paper includes 10 samples with known pathogenic expansions, which match the description provided here, but also a larger number (~100) of controls. STRetch can also be run without controls, using all samples in a batch instead, which would likely give better results than a set of only 10 controls.

We repeated the analysis twice: once utilizing all controls provided by STRetch and once without any control samples. In the latter case, we incorporated all H3Africa samples in a single

batch, as suggested by the reviewer. However, due to the distinct nature of the sequencing data for control samples in STRetch, which were PCR-free unlike the H3Africa samples, we consider the results from the second analysis to be more reliable. We updated **Supplementary Table 11** to include the results from both analyses to demonstrate whether or not STRetch supported the reported expansions by our method. All methods tested support the Africa-specific expansions at the *CA10* and *NEXN* loci.

In the Filtering initial TR genotypes subsection of the Methods: “some regions are called by both adVNTR and an STR caller...tend to have lower Mendelian error rates in adVNTR”. This is likely a typo (“lower”  “higher”)

This is correct, that should have been “higher”. This sentence has now been removed, as we have changed the filtering strategy for VNTRs to exclude these problematic “STR-like” repeats. The revised filtering strategy is described in **Methods**.

Reviewer 2 Comments for the Author...

The authors did a major effort to characterize TR in the human genome at population level. This represents the largest and most comprehensive catalog of TR sizes ever published, well beyond other efforts in this direction and it will be of great use to many researchers. Comments and suggestions:

We thank the reviewer for the encouraging comments on this work.

- limitations of the approach, related to short reads should be clearly stated

We now expand in the **Discussion** section regarding the remaining limitations for short read approaches, and reference the new TRGT tool to highlight the promise of long read approaches such as Pacbio Hifi to further improve TR catalogs.

- They validated a subset of the database calls (something like 48 sites on each of 48 samples) by capillary electrophoresis and saw largely good results, but the discrepancies were clustered on a CG-pure locus (*C9ORF72*) and heterozygous loci where they saw dropout of one allele. The authors argued that “the vast majority of TRs genotypes based on WGS are of comparable accuracy to those obtained by the experimental gold standard of fragment analysis”, but it isn't really clear to me whether this is true. Perhaps the majority of the catalog are loci that are small, non-CG-pure, and infrequently heterozygous/polymorphic. It should be discussed what percentage of the TR catalog are gc-pure loci which may be more prone to incorrect calling. If this is a very small percentage, they should discuss whether that is a biological or a technical theme. And if it is not a very small percentage, the ‘vast majority’ statement should be toned down.

We agree this was previously overstated. We revised this section to say “most” rather than the “vast majority”.

Upon further inspection of discordant calls at *C9orf72* (which are all normal range in this cohort as expected), which was the most problematic locus in the validation set, we found that four of the calls that showed the largest discrepancies were actually due to clerical errors in recording genotypes from the original Asuragen calls which have now been fixed and reflected in

Supplementary Tables 5-6 and **Supplementary Figs. 1-2**. Although we still report 9 errors at this locus, all are off by 1-2 repeats. Based on personal communication with Asuragen (and documented in PMIDs 30430876, 20056738), calls in some cases are known to differ by +/-1 repeat unit, which we believe is driving many of these errors. Although we are still likely to perform poorly at long GC-pure repeats, we do not believe this is driving the discrepancies here.

We have updated the text to mention the proportion of GC-pure repeats in our catalog (0.67%) but also now highlight another set of problematic loci, homopolymer repeats, which are both highly prevalent in our catalog (~40%) but missing from the experimental validation due to the high rate of stutter error when attempting to perform PCR-based genotyping at these loci.

- It is important to better disambiguate 'TRs' from 'their catalog of TRs'. In particular, authors need to address the fact that lots of repetitive regions in the genome exist that are big and messy and are almost certainly systematically excluded from their catalog (e.g., anything over 1kb).

Thank you for this suggestion. We have revised the text to clarify multiple statements to refer to "our catalog of TRs" and not necessarily to all TRs.

- Please state the ratio of homopolymers to all other length k-mers in the catalog. it is hard it is to tell their relative sizes.

We now mention in the first section of the results what percentage of the catalog consists of homopolymers before and after quality filtering (approximately 40% in both cases).

- The main deliverable of this manuscript is a database of TR sizes, but its difficult to access. Authors should offer a download of the allele sizes (or at least a few percentiles of sizes for each allele) as a text file either as a supplementary dataset or through their website. But also, the website is still very much a work in progress, particularly for GRCh37, so at this point I don't feel like the database is really being delivered to readers in any effective way.

We have now significantly updated our web resource, webstr.ucsd.edu, which now allows users to browse this callset (which is based on GRCh38), as well as previous TR callsets we had released for hg19/GRCh37. WebSTR allows users to browse by region or gene, and view details including population-specific allele frequencies for each repeat. We additionally provide downloadable text files with population-specific allele frequencies for each VCF. Finally, raw data in VCF format is available on the EnsembleTR website.

We now provide links to all of these resources in the **Data Availability** section.

- It is unclear what result is returned in this catalog if a locus has a motif change that is not explicitly encoded in a tool like ExpansionHunter (like if there is a locus similar to RFC1 that hasn't been well-studied yet). They make a lot of arguments that HipSTR handles that, but only if it is below ~150bp, correct? What happens if the allele is larger?

HipSTR is not able to genotype regions with lengths above 150bp. So in those cases, we won't be able to say anything about the sequence imperfection. In shorter cases, HipSTR is able to resolve repeat unit changes. We now provide examples at the CA10 and NEXN TRs, in which we identified population-specific expansions and alleles with high rates of sequence impurity (new **Supplementary Fig. 14**) to make this more clear.

If these motif changes occur at very long repeats that cannot be genotyped by HipSTR (such as pathogenic expansions in *RFC1*), and the repeat structure was not known beforehand, genotypes in our catalog will not capture the sequence change although often still return reasonable length estimates. For example, we identified cases of long alleles at *NEXN* genotyped by ExpansionHunter where an expansion was correctly identified but reported as a perfect repeat, but where motif changes were clearly evident from manual inspection of Pacbio hifi reads. We further emphasize in the **Discussion** that further work is needed to resolve these complex cases, and cite the new TRGT approach as a promising direction to do so.

- the explanation of the classifier performance in the 'sequence determinants of TR polymorphism' section is difficult to interpret. Due to the probable class imbalance, authors need to give quite a bit more detail here --actual accuracy, precision and recall values rather than just F1-scores, as well as area under the precision-recall curves would be nice.

We now report these metrics in the text. Our model achieved an overall accuracy of 73% on a held out test set with area under the precision recall curve 0.82/0.62 when considering stable/polymorphic STRs as the target class (precision=0.76/0.64 for stable/polymorphic and recall=0.87/0.46 for stable/polymorphic).

- more generally, does this 4-mer observation explain a subset of sites well or a large number of sites weakly? I would appreciate some numbers of what percentage of their stable vs polymorphic loci exhibited this pattern. The quantification with spearman is nice, but doesn't really tell a lot.

We have expanded this description to try and address this question. Generally, the 4-mer observation seems to explain a subset of sites well. While nearly all STRs contain at least one of these 4mers in their context, STRs with higher counts of these repeat-like 4mers tend to be much more polymorphic. We quantify this by dividing the counts of these 4mers into quintiles. We find polymorphic STRs are highly prevalent in the upper quintile (27.6% of polymorphic STRs fall into this bin, compared to 15.5% of stable STRs) and depleted in the lowest quintile (which contains 16.1% of polymorphic STRs and 23.1% of stable STRs), whereas the middle quintiles have similar representation of stable vs. polymorphic repeats.

- then utility of imputation TRs with nearby SNPs is somewhat overstated. As expected this is largely confined to stable TRs. The medically (GWAS) most interesting TRs are the unstable ones, especially the rare expansions. There is not a god reason to believe that nearby SNPs would capture this well (other than in founder haplotypes). Also given the meiotic instability and dynamism of expansions. This should be made more clear. I am not convinced (yet) that this is a true advance for GWAS studies.

The reviewer is correct that the imputability of a TR is highly dependent on how stable the TR is. Highly polymorphic loci, such as those implicated in repeat expansion disorders such as Huntington's Disease, are particularly challenging.

However, the majority of TRs can actually be computed quite well, and the ability to impute TRs has already provided a clear advance for applications in GWAS, as evidenced by studies from us and other groups. In our own work (<https://www.biorxiv.org/content/10.1101/2022.08.01.502370v1>), we recently imputed TRs into UK Biobank using our previous haplotype panel based largely on European samples and tested

for association with blood traits. After stringent fine-mapping, we identified 118 high confidence putatively causal TR-trait associations. For example, we identified a highly polymorphic CGG repeat in the promoter of *CBL* as a driver of platelet related traits and a poly-Serine coding repeat in *E2F4* strongly associated with many red blood cell related traits. Further, using a different imputation approach Mukamel et al. recently imputed VNTRs into UKBiobank which identified multiple high effect TRs likely to be causal drivers of the strongest GWAS signals for height and other traits (PMID: 34554798). We now highlight this further in the **Discussion**.

We have now expanded the section in our manuscript on imputation to include a table of imputation accuracy metrics at medically relevant TRs (new **Supplementary Table 17**), including TRs implicated in Mendelian disorders as well as candidate causal TRs identified in our GWAS work (Margoliash et al.) to demonstrate the feasibility of using imputation to detect medically relevant TR-trait associations.

REVIEWERS' COMMENTS

Reviewer #1 (Remarks to the Author):

The authors have provided a careful and thorough response to the points raised. I am satisfied with their responses and feel that the manuscript is substantially improved.

Minor comment on the new Table S1: please add a description of what the number in the brackets represents (number of TR calls from which the MI is determined?).

One issue related to the newly added section on imputation of medically relevant TRs and in particular the statement "...indicating that a range of medically relevant repeats can be accurately imputed". For the expansion disorders, in particular, it is large expansions that are medically relevant however these are rare and I assume are not present in the reference panel. Imputation performance was only evaluated in terms of average concordance, so it is mainly just evaluating imputation accuracy for the common allele sizes. Greater context should be provided here about the range of allele sizes in the panel and evaluation and therefore which allele size might be expected to be imputed with reasonable accuracy and which would not.

Reviewer #2 (Remarks to the Author):

The detailed response addresses all my concerns.

REVIEWERS' COMMENTS

Reviewer #1 (Remarks to the Author):

The authors have provided a careful and thorough response to the points raised. I am satisfied with their responses and feel that the manuscript is substantially improved.

Minor comment on the new Table S1: please add a description of what the number in the brackets represents (number of TR calls from which the MI is determined?).

Thank you for mentioning this. Yes, numbers show the number of trio/TR pairs that were analyzed for each combination of tools.

One issue related to the newly added section on imputation of medically relevant TRs and in particular the statement "...indicating that a range of medically relevant repeats can be accurately imputed". For the expansion disorders, in particular, it is large expansions that are medically relevant however these are rare and I assume are not present in the reference panel. Imputation performance was only evaluated in terms of average concordance, so it is mainly just evaluating imputation accuracy for the common allele sizes. Greater context should be provided here about the range of allele sizes in the panel and evaluation and therefore which allele size might be expected to be imputed with reasonable accuracy and which would not.

We have clarified in this section that only common, normal length alleles are included in our panel. Specifically, we added the sentence:

"Notably, our panel is comprised of healthy controls, and thus imputation is restricted to common normal range alleles for the known pathogenic loci."

And edited the following sentence (new text in red):

"Overall the average concordance across all pathogenic loci remained high (range 0.89-0.92 across populations) indicating that common, non-pathogenic alleles at a range of medically relevant repeats can be accurately imputed."

Reviewer #2 (Remarks to the Author):

The detailed response addresses all my concerns.